# LP-RAG: A Link Prediction-Based Framework for Retrieval-Augmented Generation

## Abstract

Retrieval-augmented generation (RAG) strategies have empowered large language models (LLMs) through integration with external knowledge sources, thereby enabling more accurate, up-to-date, and contextually relevant outputs. Among these, graph-based RAG methods stand out as particularly prominent. These approaches aim to structure external knowledge into graphs and leverage relational reasoning to retrieve relevant information to the context at hand. However, existing approaches remain limited in their ability to exploit query-based semantic cues. In this paper, we propose LP-RAG, a link prediction-based framework for document RAG. Specifically, LP-RAG employs an LLM-prompted chunker and text encoders to construct a graph of similarity relationships among chunks, which is then augmented with chunk-conditioned synthetic queries that emulate potential questions for each chunk. This design enables the incorporation of chunk-specific semantic information for model training. In our framework, retrieval is cast as an inductive link prediction problem, where the goal is to predict chunk–query links. Notably, LP-RAG is model-agnostic and can incorporate any link prediction method (e.g., graph neural network–based predictors). To demonstrate its effectiveness, we evaluate LP-RAG across diverse settings/benchmarks. Results show that LP-RAG consistently outperforms existing graph-based RAG methods.

## 1 Introduction

Large language models (LLMs) (Brown et al., 2020; Chowdhery et al., 2023; Kaplan et al., 2020) have quickly become one of the most transformative technologies of modern era, enabling significant progress across a range of applications, such as conversational agents (Pan et al., 2025), automated code generation (Liu et al., 2023), education (Liu et al., 2024), and healthcare (Kim et al., 2024; Wu et al., 2024). Yet, their reliance on static corpora prevents access to real-time information, limits coverage of domain-specific knowledge, and hinders reliable encoding of long-tail factual knowledge (Kandpal et al., 2023; Luo et al., 2025). To address these limitations, retrieval-augmented generation (RAG) systems (Lewis et al., 2020; Jimenez Gutierrez et al., 2024; Xu et al., 2025) couple LLMs with information retrieval strategies, integrating external knowledge into the generation process. Nowadays, RAG systems constitute a central component of LLM-based applications, underpinning widely deployed products such as search engines and chat assistants (Cai et al., 2024).

Despite rapid progress, most RAG methods still lack a high-level, global understanding of knowledge distributed across documents and do not provide principled mechanisms to exploit relationships among their textual elements (Luo et al., 2024; Xu et al., 2025). As a result, they often struggle with multi-hop reasoning and summary-level queries (Han et al., 2024b). To address these limitations, graph-based RAG (GraphRAG) (Han et al., 2023; Jimenez Gutierrez et al., 2024) emerged as an alternative that aim to organize the text corpus into a graph, where nodes typically represent entities or chunks and edges encode diverse relation types. Recent approaches further leverage LLMs to incorporate summary-based information into graphs (Edge et al., 2024; Jimenez Gutierrez et al., 2024), and employ graph neural networks (GNNs) (Scarselli et al., 2009; Gilmer et al., 2017) to reason over these structures (Mavromatis & Karypis, 2025; He et al., 2024).

However, we argue that existing approaches overlook query-based semantic cues for retrieval tasks. To the best of our knowledge, no prior works leverage simulated queries as supervision signal for learnable retrieval methods on graph data. This is critical since simulated queries provide a direct

proxy for the information needs of downstream users, enabling the model to align graph traversal and context selection with query intent rather than relying solely on static graph connectivity.

In this paper, we propose LP-RAG — a novel link prediction-based approach for QA document RAG. Our key insight is to leverage LLM-prompted synthetic queries to augment graph indexes with fine-grained semantic information. Importantly, our design enables $i$) exploiting semantic cues from chunks (candidate passages for retrieval), and $ii$) integrating learned powerful link prediction methods for retrieval. More specifically, our methodology begins by breaking down complex and abstract concepts into simpler chunks using a prompt-driven strategy with a small-scale language model. Each resulting chunk is represented as a node, and connections between nodes are established through similarity scores from

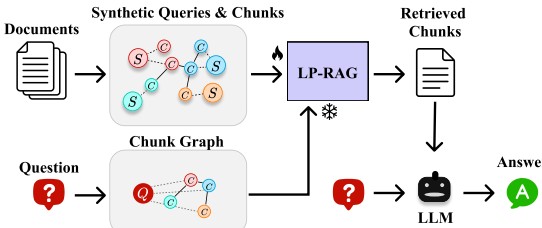

Figure 1: **Overview of LP-RAG**. We use the chunk-query graph for training. For inference, we feed only the chunk graph and the user query to LP-RAG to obtain chunks for generation.

contextual embeddings. Subsequently, synthetic queries are generated by the language model conditioned on the extracted chunks. These chunks, queries, and inter-node connections are then integrated to form a unified graph structure, referred to as the chunk-query graph. Finally, we formulate retrieval as a link prediction problem, where the goal is to predict links between an incoming query and chunk nodes. Crucially, the use of synthetic queries provides direct supervision for training the link predictors, enabling LP-RAG to flexibly incorporate a wide range of link prediction models. Figure 1 illustrates the steps of LP-RAG.

To evaluate the effectiveness of LP-RAG, we mainly follow the experimental setups of Xu et al. (2025) and Luo et al. (2025) and consider ten multi-hop RAG benchmarks. Our results demonstrate that LP-RAG consistently outperforms strong baselines, including NodeRAG, HippoRAG, and GFM-RAG, in open-ended head-to-head evaluations, while maintaining a competitive number of retrieval tokens. We also conduct ablation studies to demonstrate the importance of using synthetic queries in our framework, and the impact of different GNNs and negative sampling schemes.

In sum, our contributions are:

- We are the first to formulate RAG as a link prediction problem, enabling us to leverage the rich available toolbox of link prediction methods;
- We propose LP-RAG, a simple graph-based RAG approach that integrates synthetic queries, LLM-prompted chunker, and link predictors for effective retrieval on graphs;
- We demonstrate the efficiency and effectiveness of LP-RAG in multiple domains, with significant empirical gains over recently introduced baselines, like NodeRAG and GFM-RAG.

## 2 RELATED WORKS

**Retrieval-Augmented Generation (RAG)** (Lewis et al., 2020; Gao et al., 2023b) provides an effective way to integrate external knowledge into LLMs by retrieving relevant documents to facilitate LLM generation. Early works adopt the pre-trained dense embedding model to encode documents as separate vectors (Lewis et al., 2020; Chen et al., 2024a; Moreira et al., 2024), which are then retrieved via similarity scores to the query. Despite efficiency and generalizability, these methods struggle to capture complex document relationships. Subsequent studies have explored multi-step retrieval, where LLMs guide an iterative process to retrieve and reason over multiple documents (Gao et al., 2023a; Su et al., 2024).

**Graph-based RAG** (Peng et al., 2024; Han et al., 2024a) leverages graphs to explicitly model the complex relationships between knowledge, facilitating retrieval and reasoning. The methods GraphRAG (Han et al., 2023; Edge et al., 2024) and LightRAG (Guo et al., 2025) incorporate graph structures into text indexing and retrieval, enabling efficient retrieval of entities and their relationships. HippoRAG (Jimenez Gutierrez et al., 2024) enhances multi-hop retrieval by using a personalized PageRank algorithm to locate relevant knowledge. NodeRAG (Xu et al., 2025) further

advances this line of work by introducing heterogeneous graph structures specifically designed to enable a more seamless and holistic integration of graph algorithms into the RAG workflow. However, the graph structure can be noisy and incomplete, leading to suboptimal performance. Efforts to incorporate GNNs into graph-enhanced RAG (He et al., 2024; Mavromatis & Karypis, 2025; Li et al., 2025b; Luo et al., 2025) have shown impressive results due to the strong graph reasoning capabilities of GNNs in handling graph data. Nonetheless, these methods still limit in generalizability due to the dependence, in some degree, on similarity algorithms during the retrieval stage.

**Knowledge graph construction & reasoning.** Other frameworks focus on constructing and fusing structured knowledge: RAGraph retrieves whole graphs from a library to augment downstream graph learning (Jiang et al., 2024), while Graphusion uses LLMs to extract triplets and fuse them into a global knowledge graph for improved retrieval and reasoning (Yang et al., 2025). TOBU-Graph (Kashmira et al., 2024) instead builds knowledge graphs from documents and perform graph-traversal retrieval on these KGs, providing an alternative to text-only similarity retrieval Kashmira et al. (2024). Finally, specialized methods target schema-matching and multi-hop retrieval by combining vector-based matching with graph-traversal over large external KGs (Ma et al., 2025).

**Link prediction** aims to infer missing or future edges in a graph by estimating a score for each candidate node pair. Classical structural heuristics — e.g., Common Neighbors, Jaccard and Adamic–Adar — remain strong, scalable baselines and are commonly used for social and information networks (Liben-Nowell & Kleinberg, 2003; Yao et al., 2016; Adamic & Adar, 2003). More recently, neural methods have dominated benchmarks: variational graph autoencoders learn node latents specifically for reconstructing edges (Kipf & Welling, 2016), while subgraph-centric GNN methods, such as SEAL, extract enclosing subgraphs and learn structure-aware classifiers (Zhang & Chen, 2018). Equivariant positional-encoding GNNs (e.g., PEG, Wang et al. (2022)) and Neural Common Neighbor (NCN) (Wang et al., 2024) architectures enhance expressivity by combining structural features and MPNN representations.

Collectively, these graph-based RAG methods emphasize graph construction (passage graphs or knowledge graphs) and the use of graph traversal or GNN-based ranking to guide retrieval (Peng et al., 2024). To the best of our knowledge, framing retrieval as a supervised link-prediction problem — where chunk and query nodes are explicitly linked and edge existence is learned — remains unexplored. Our proposed LP-RAG fills this gap by (i) decomposing documents into chunk-nodes and synthetic query-nodes, and (ii) learning a link-prediction model to score query–chunk edges.

## 3 METHOD: LP-RAG

This section introduces LP-RAG, a novel link prediction–based framework for retrieval-augmented generation. We begin with a brief overview of the document RAG task and our approach, and then Sections 3.1–3.4 provide a detailed description of each component of LP-RAG. A detailed example of the LP-RAG process can be found in Appendix A.

**Notation.** A corpus is a collection of documents $\mathcal{D} = \{D_1, \ldots, D_{|\mathcal{D}|}\}$, where each document $D_i$ is represented as a sequence of tokens $(x_1^i, \ldots, x_{|D_i|}^i)$. A *chunk* is defined as a contiguous subsequence of tokens within a document. Formally, a chunk $x_{s:e}^i$ corresponds to the slice $(x_s^i, \ldots, x_e^i)$ with $1 \leq s \leq e \leq |D_i|$. Moreover, we represent an *attributed undirected graph* as a tuple $G = (V, E, z)$, where $V$ is a set of vertices, $E$ is a set of unordered pairs of vertices, called edges, and $z : V \to \mathbb{R}^d$ is an attribute function that assigns a feature (or color) $z(v)$ to each vertex $v \in V$. The set of neighbors of a node $v$ in $G$ is denoted by $\mathcal{N}^G(v) = \{u \in V : \{v, u\} \in E\}$. Hereafter, we denote the features of $v$ by $z_v$.

**Document RAG tasks.** Let $\mathcal{C}$ denote the collection of all chunks. Given a user query $q$, a corpus $\mathcal{D}$, and a pre-trained language model $\text{LLM}(\cdot)$, we want to find a retrieval function RETRIEVE, with

$$\mathcal{C}_q = \text{RETRIEVE}(q, \mathcal{D}) \subseteq \mathcal{C}, \tag{1}$$

such that the retrieved chunks $\mathcal{C}_q$ are relevant to $q$. The notion of relevance is broad and task-dependent. Here, we are particularly interested in retrieval functions for which the combination of $q$ and $\mathcal{C}_q$, denoted by $\text{COMBINE}(q, \mathcal{C}_q)$, maximizes the expected utility of the LLM answers

$$y_q \sim \text{LLM}\big(\text{COMBINE}(q, \mathcal{C}_q)\big). \tag{2}$$

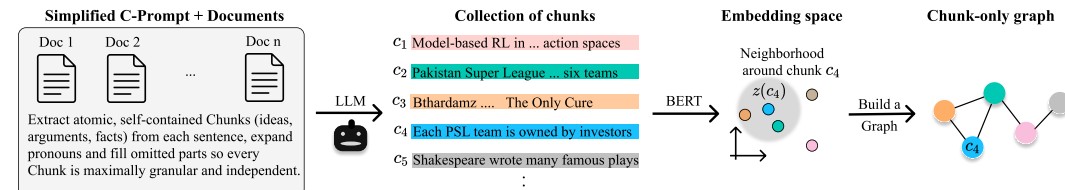

Figure 2: **Chunking**. LP-RAG decomposes a document into chunks via a prompted LLM. Then, we compute embeddings for each chunk node using a BERT-based encoder. Next, we calculate pairwise cosine similarity scores between all chunk-node embeddings. Finally, each chunk node is connected to its top-$k$ most similar neighbors, as given by these similarity scores.

The notion of utility is task-dependent, e.g., semantic equivalence or exact match for Q&A. Equations 1 and 2 denote the retrieval and generation phases of RAG systems, respectively.

**Overview.** Our approach targets the retrieval stage by representing the corpus with a graph index. In LP-RAG, we first construct a graph whose nodes are either document chunks or synthetically generated queries, and whose edges encode node-embedding similarity. The key idea is that synthetic queries expose chunk-specific semantics that chunk-only graphs may miss. We then train a general link-prediction model on this synthetic graph to capture chunk–query relationships. At inference, given a new query $q$, retrieval reduces to predicting links from $q$ to chunk nodes and ranking the top-scored candidates. We detail each step below and provide an overview of LP-RAG in Figure 1.

### 3.1 CHUNK EXTRACTION

A key challenge in applying RAG to augment LLMs is document chunking. Chunks must balance granularity and coverage: they should be small enough to enable efficient retrieval and stay within the model's input token budget, yet sufficiently informative to remain semantically coherent and contain the evidence needed to answer the user query.

We carry out chunking via a prompted LLM. The idea is that leveraging prompted LLMs for chunk extraction help to mitigate truncated content and per-chunk ambiguity while obviating manual tuning of chunk length. It may also alleviate overlap issues commonly observed in heuristic approaches (e.g., TextSplit, SemanticSplit, and RecursiveSplit) (Lensu, 2025; Stäbler et al., 2025).

Let C-PROMPT denote a fixed instruction prompt (for details, see Appendix C). We apply an LLM, conditioned on C-PROMPT and the corpus $\mathcal{D}$, to obtain the collection of chunks, i.e., the chunk set is $\mathcal{C} = \mathrm{LLM}(\text{C-PROMPT} \,\|\, \mathcal{D})$. A illustration of this procedure is given in Figure 2.

### 3.2 SYNTHETIC QUERY GENERATION

We prompt an LLM to generate synthetic queries conditioned on the extracted chunks. The goal is for the LLM to produce queries that closely resemble those posed by human users.

Let $C \in \mathcal{C}$ be a target chunk. We prompt an LLM to generate synthetic queries whose answers are given in $C$. Using a fixed prompt S-PROMPT (see Appendix C for a detailed description), we generate up to $n_C$ synthetic queries for $C$:

$$\mathcal{S}_C \sim \mathrm{LLM}(\text{S-PROMPT} \,\|\, C; n_C). \tag{3}$$

We denote the full set of synthetic queries by $\mathcal{S} = \bigcup_{C \in \mathcal{C}} \mathcal{S}_C$.

For efficiency, we employ batched generation: given a batch $\mathcal{B} \subseteq \mathcal{C}$, the LLM is prompted jointly to produce $\mathcal{S}_\mathcal{B} = \bigcup_{C \in \mathcal{B}} \mathcal{S}_C$. Note that a synthetic query can be associated with one or more chunks. We record chunk–query associations via the relation $\mathcal{A} \subseteq \mathcal{S} \times \mathcal{C}$, with $(S, C) \in \mathcal{A}$ for every $S \in \mathcal{S}_C$.

### 3.3 GRAPH CONSTRUCTION

Given the chunk set $\mathcal{C}$ and the synthetic query set $\mathcal{S}$, we build an undirected attributed graph $G = (V, E, z)$ with node set $V = \mathcal{C} \cup \mathcal{S}$, node-feature function $z$ given by a pre-trained contextual encoder (e.g., Contriever (Izacard et al., 2022)), and edge set $E$ defined as follows.

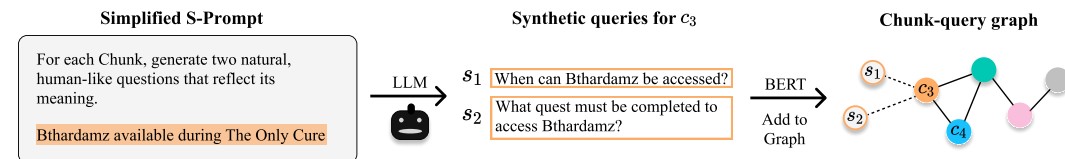

Figure 3: **Synthetic queries and chunk-query graphs**. For each chunk node, we create some synthetic queries via LLMs. We convert the synthetic queries into embeddings using Contriever. Finally, each synthetic query is added to the graph and connected to its respective chunks.

**Query-chunk edges.** Let $\mathcal{A} \subseteq \mathcal{S} \times \mathcal{C}$ denote the query-chunk relation (Section 3.2). We add an edge between a query $S$ and a chunk $C$ if, and only if, $(S, C) \in \mathcal{A}$, i.e.,

$$\{S, C\} \in E \iff (S, C) \in \mathcal{A} \quad \text{(equivalently, } S \in \mathcal{S}_C\text{)}. \tag{4}$$

**Chunk-chunk edges.** Let $\text{sim} : \mathbb{R}^d \times \mathbb{R}^d \to \mathbb{R}$ be a similarity metric (e.g., cosine similarity) in the embedding space. Also, for $k \in \mathbb{N}$, define the $k$-NN set of a chunk $C$ as

$$\mathcal{N}_k(C) = \arg \text{topk}\{\text{sim}\left(z_{C'}, z_C\right) : C' \in \mathcal{C} \setminus \{C\}\}.$$

Then, we connect two chunks $C$ and $C'$ by *mutual $k$-NN*, that is

$$\{C, C'\} \in E \iff C \in \mathcal{N}_k(C') \land C' \in \mathcal{N}_k(C). \tag{5}$$

Regarding scalability, we note that, with embeddings cached and an approximate nearest-neighbor (ANN) index over $\{z_C\}_{C \in \mathcal{C}}$, adding a batch of new chunks requires $O(|\mathcal{B}|)$ encoder calls and $O(|\mathcal{B}| \log |\mathcal{C}|)$ (amortized) neighbor lookups, yielding incremental, near-linear updates in batch size.

### 3.4 LEARNING TO RETRIEVE

We now formulate retrieval as an inductive link prediction task. Let $G' = (V', E')$ be an (unknown) underlying graph and let $G = (V, E)$ with $V \subseteq V'$ and $E \subseteq E'$ be an observed subgraph. In standard link prediction, the goal is to infer missing edges in $E' \setminus E$ among nodes in $V$. In the *inductive* setting, the predictor must also infer edges involving unseen nodes in $V' \setminus V$. In our context, the observed graph corresponds to the attributed query-chunk graph $G$ (in Section 3.2) and the goal is to predict links between incoming user queries (treated as unseen nodes) and chunk nodes in $G$.

Formally, we consider parameterized link predictors of the form $\phi_\theta : V \times V \to \mathbb{R}$, with parameters $\theta \in \mathbb{R}^m$. Given the undirected nature of our graphs, we further require $\phi_\theta(u, v) = \phi_\theta(v, u) \, \forall \, u, v$. For an unseen query $q \notin V$, we create the augmented graph $\tilde{G} = (V \cup \{q\}, E)$ and apply $\phi_\theta$ inductively to pairs of nodes in $\tilde{G}$. Then, retrieval amounts to scoring candidate edges between $q$ and chunk nodes $C \in \mathcal{C}$ and selecting those with the highest values:

$$\mathcal{C}_q = \arg \text{topk}\{\phi_\theta(q, C) : C \in \mathcal{C}\} \quad \text{or} \quad \mathcal{C}_q = \{C \in \mathcal{C} : \phi_\theta(q, C) \geq \tau\}, \tag{6}$$

for a chosen $k$ or threshold $\tau$. Therefore, the retrieved set $\mathcal{C}_q$ comprises the chunk nodes predicted to form links with $q$.

For simplicity, we have omitted node features from this formulation. However, we recall that each node is endowed with a contextual embedding produced by a text encoder (e.g., BERT; see Section 3.3). We also highlight that our approach is rather general and accommodate arbitrary models. Box 3.4 briefly illustrates the use of GNNs for link prediction.

Regarding training, as usual, we apply the binary cross-entropy (BCE) loss with negative sampling to learn the parameters $\theta$ of our predictor $\phi_\theta$. Let us define $\hat{y}_{u,v} = \sigma(\phi_\theta(u, v))$ where $\sigma$ denotes the sigmoid logistic function. Then, we want to minimize the loss

$$\mathcal{L} = - \sum_{\{u,v\} \in E} \left(\log \hat{y}_{u,v} + \mathbb{E}_{v' \sim \mathbb{P}_V(u)}[\log(1 - \hat{y}_{u,v'})]\right), \tag{10}$$

---

**Box 3.4: Graph Neural Networks (GNNs) for Link Prediction**

Most GNNs can be described using the message-passing paradigm (Gilmer et al., 2017). They employ a sequence of message-passing steps, where each node $v$ aggregates messages from its neighbors $\mathcal{N}^G(v)$ and use the resulting vector to update its own embedding.

Let $h_v^{(0)} = z_v$ for all $v \in V$. Message-passing GNNs recursively compute

$$m_v^{(\ell)} = f_{\text{agg}}^{(\ell)} \left( \{\!\{ h_u^{(\ell-1)} : u \in \mathcal{N}(v) \}\!\} \right) \quad \forall v \in V \tag{7}$$

$$h_v^{(\ell)} = f_{\text{upd}}^{(\ell)}(h_v^{(\ell-1)}, m_v^{(\ell)}) \quad \forall v \in V, \tag{8}$$

where $f_{\text{agg}}$ is an order-invariant function and $f_{\text{upd}}$ is an arbitrary map (often an MLP). The final GNN embeddings, $\{h_v^{(L)}\}_{v \in V}$, can be further processed using feedforward layers. Finally, the score for a link $\{u, v\}$ is given by

$$\phi(u, v) = \text{MLP} \circ \text{Readout}(\{h_v^{(L)}, h_u^{(L)}\}). \tag{9}$$

---

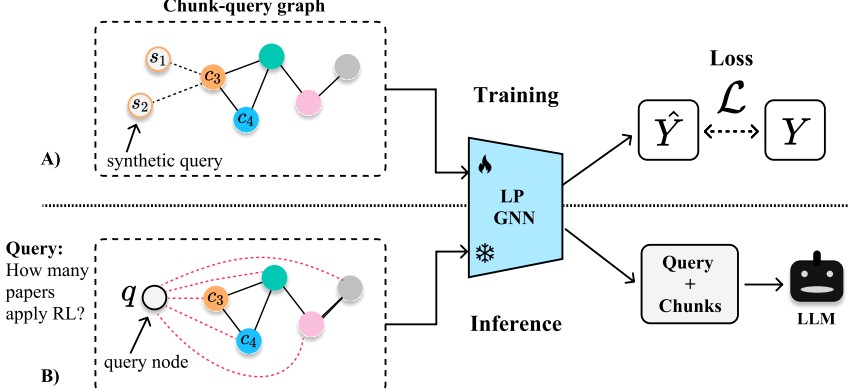

Figure 4: A) **GNN training** with the graph formed by the method. The edges connecting the queries are given greater importance; B) For **inference**, synthetic query nodes are removed from the graph.

where $\mathbb{P}_V(u)$ is a negative-sampling distribution over $V$. In practice, we employ Monte-Carlo approximation $\mathcal{L} = -\sum_{\{u,v\} \in E}(\log \hat{y}_{u,v} + |P(u)|^{-1} \sum_{v' \in P(u)} \log(1 - \hat{y}_{u,v'}))$, where $P(u)$ is a small set of i.i.d. samples from $\mathbb{P}_V(u)$. Figure 4 shows the training/test phases of GNN-based LP-RAG.

**Integration into LLMs.** The final step of LP-RAG consists of combining the input user prompt with the retrieved chunks. Given the retrieved set $\mathcal{C}_q$, we sort the chunks in reverse order by their link–prediction scores w.r.t. $q$, i.e., we obtain a sequence of chunks $C_1, C_2, \ldots, C_{|\mathcal{C}_q|} \in \mathcal{C}_q$ such that $\phi_\theta(q, C_i) \geq \phi_\theta(q, C_{i+1})$ for all $i$. We then concatenate the query with the sorted chunks to form the input to the LLM:

$$\text{COMBINE}(q, \mathcal{C}_q) = q \,\|\, C_1 \,\|\, C_2 \,\|\, \cdots \,\|\, C_m.$$

The exact final prompt sent to the LLM is described in Appendix C.

## 4 EXPERIMENTS

This section evaluates LP-RAG on widely used benchmarks. We conduct three sets of experiments designed to assess our approach across diverse conditions, including different LLMs, data scales, and evaluation benchmarks. We also perform ablation studies to show the contribution of different design choices: the effect of diverse link prediction methods, the benefit of synthetic queries within LP-RAG, and the impact of different negative sampling strategies. Implementation details are provided in Appendix B, and Appendix C contains the set of prompts used in our experiments.

Table 1: Q&A results: LP-RAG consistently outperforms all baselines in all datasets while utilizing minimum resources. Bold text denotes the best results.

| Methods | HotpotQA-S | | | MuSiQue-S | | | MultiHop-S | | | RAG-QA-S | | | ICLR-S | | |
|---|---|---|---|---|---|---|---|---|---|---|---|---|---|---|---|
| | Acc.↑ | F1↑ | Tok↓ | Acc.↑ | F1↑ | Tok↓ | Acc.↑ | F1↑ | Tok↓ | Acc.↑ | F1↑ | Tok↓ | Acc.↑ | F1↑ | Tok↓ |
| NaiveRAG | 67.50 | 22.50 | 7.2k | 69.43 | 20.30 | 7.1k | 68.23 | 21.14 | 8.1k | 67.08 | 25.50 | 8.2k | 53.90 | 21.62 | 7.9k |
| HyDe | 72.17 | 36.95 | 7.3k | 69.43 | 34.60 | 6.7k | 73.89 | 37.59 | 8.3k | 54.96 | 38.63 | 8.4k | 68.50 | 35.30 | 7.2k |
| GraphRAG | 87.24 | 65.00 | 6.1k | 61.71 | 42.00 | 5.8k | 72.15 | 58.62 | 6.4k | 65.96 | 43.60 | 6.4k | 74.50 | 42.00 | 6.9k |
| LightRAG | 79.83 | 65.00 | 5.4k | 63.14 | 41.50 | 5.3k | 70.63 | 48.95 | 6.7k | 65.89 | 43.62 | 6.3k | 72.40 | 43.96 | 6.9k |
| NodeRAG | 86.90 | 69.50 | 4.0k | 66.00 | 45.00 | 5.2k | 70.86 | 54.50 | 6.0k | 67.20 | 56.86 | 3.3k | 77.92 | 54.82 | 6.2k |
| LP-RAG | **89.32** | **86.29** | **3.2k** | **89.67** | **85.15** | **3.8k** | **89.12** | **86.35** | **3.8k** | **87.89** | **85.26** | **2.2k** | **91.16** | **89.51** | **2.4k** |

## 4.1 EXPERIMENT I: OPEN-SOURCE LLM IN SMALL-DATA REGIMES

In this experiment, we evaluate NaiveRAG (Lewis et al., 2020), HyDe (Gao et al., 2023a), GraphRAG (Edge et al., 2024), LightRAG (Guo et al., 2025), NodeRAG (Xu et al., 2025), and LP-RAG (ours) across smaller versions of five popular benchmarks: HotpotQA (Yang et al., 2018), MuSiQue (Trivedi et al., 2022), MultiHop-RAG (Tang & Yang, 2024), RAG-QA (Han et al., 2024b), and ICLR Papers (Feng et al., 2025). Following Xu et al. (2025), we compute macro accuracy, F1 score, and measure the average number of retrieved tokens to evaluate efficiency.

**Implementation details.** We conduct a controlled experiment using `LLaMA 3.3-70B` and sampling 20 documents per dataset — except for ICLR Papers, for which we sample 50 documents. To distinguish the reduced datasets from their original counterparts, we append the suffix "-S" to the smaller versions. This reduced setting enabled us to (1) re-run baselines and perform ablation studies, and (2) compute F1 scores (not available in Xu et al. (2025)), ensuring a fair and reliable comparison. Link prediction methods were implemented using PyTorch / PyTorch Geometric (Fey & Lenssen, 2019). As link predictor, we employ the NCN model (Wang et al., 2024). Additionally, we use Contriever (Izacard et al., 2022) as embedding model for all texts.

Regarding synthetic queries, we generate two queries per chunk. We observed that LLMs tend to produce highly similar queries when prompted for multiple variants; hence, we additionally generate queries spanning connected chunks. This step improves the model's ability to handle 2-, 3-, and 4-hop reasoning queries. We provide further details in Appendix B.

**Results.** Table 1 shows the results. Across all datasets, LP-RAG significantly outperforms all baselines: for the HotpotQA-S dataset, it achieves a 22% accuracy advantage over NaiveRAG, and a 2.42% over NodeRAG. Furthermore, its cost is lower than that of all baselines across datasets, with particularly large savings on the ICLR dataset. Regarding similarity-based baselines (i.e., NaiveRAG, HyDe, and GraphRAG), they generally achieve lower accuracy and macro F1 scores, even when they use graphs to capture relational structure. This behavior is evidenced in the GraphRAG results where accuracy is lower than NaiveRAG in the MuSiQue-S dataset. These findings suggest that these similarity-based methods struggle to identify the most informative nodes for generation, reinforcing the benefits of LP-RAG's link-prediction-based retrieval mechanism.

We hypothesize that LP-RAG achieves the lowest token consumption for two reasons. First, our chunk extractor produces short, atomic chunks. Second, retrieval is based on the probability of an edge existing in a chunk rather than a fixed top-$k$ selection. Together, these factors yield higher-quality information and more efficient retrieval. Overall, our results suggest that link prediction methods can improve retrieval performance in LLM-based question-answering tasks.

## 4.2 EXPERIMENT II: PRIVATE LLM ON ARENA BENCHMARKS

In this experiment, we compare LP-RAG against the same baselines of EXPERIMENT I using a private LLM on larger datasets (thousands of documents). Following the evaluation protocol of Xu et al. (2025), we adopt `GPT-4.o-mini` as LLM and use the Arena Benchmarks (Han et al., 2024b).

For fairness, we use the exact data splits of Xu et al. (2025) and consider four domains: Writing, Recreation, Science, and Tech, which keeps evaluation costs manageable. Baseline results are taken directly from their work. Consistent with their evaluation, we measure token usage for efficiency and report the win–tie ratio (W+T). For LP-RAG, we use the same configuration described in Section 4.1.

Table 2: Results on the Arena datasets. LP-RAG surpasses the performance of the baselines in 2 out of 4 datasets. Best values are in bold.

| Methods | Writing | | Tech | | Science | | Recreation | |
|---|---|---|---|---|---|---|---|---|
| | W+T ↑ | Token↓ | W+T ↑ | Token↓ | W+T ↑ | Token↓ | W+T ↑ | Token↓ |
| NaiveRAG | 0.663 | 9.4k | 0.689 | 9.1k | 0.526 | 9.0k | 0.720 | 9.1k |
| HyDe | 0.789 | 9.6k | 0.863 | 9.3k | 0.823 | 9.3k | 0.777 | 9.3k |
| LightRAG | 0.754 | 6.3k | 0.937 | 6.9k | 0.840 | 7.1k | 0.800 | 6.8k |
| GraphRAG | 0.749 | 6.4k | 0.943 | 6.7k | 0.863 | 6.7k | 0.806 | 6.8k |
| NodeRAG | **0.794** | 3.3k | **0.949** | 3.8k | 0.903 | 4.2k | 0.886 | 3.3k |
| LP-RAG | 0.775 | 4.2k | 0.948 | 4.8k | **0.920** | 4.1k | **0.907** | 4.1k |

**Results.** Table 2 reports the results on the Arena benchmarks. LP-RAG attains the highest win/tie ratio in both the Science and Recreation domains. It outperforms NaiveRAG, HyDe, LightRAG, and GraphRAG in three domains (Science, Recreation, and Technology), with particularly large gains in Science, where it exceeds NaiveRAG by 0.394. In the Writing domain, LP-RAG does not surpass NodeRAG or HyDe, but it still improves over GraphRAG by 0.026 while using substantially fewer tokens. Overall, LP-RAG's token usage remains comparable to NodeRAG across all domains.

## 4.3 EXPERIMENT III: ADDITIONAL COMPARISON ON STANDARD BENCHMARKS

Now we evaluate the retrieval and the question-answering performance of LP-RAG against several methods. More specifically, we follow the same setup by Guo et al. (2025), which includes: three widely-used multi-hop QA datasets, i.e., HotpotQA (Yang et al., 2018), MuSiQue (Trivedi et al., 2022), and 2WikiMultiHopQA (2Wiki) (Ho et al., 2020); a well-documented separation of data to avoid data leakage; two evaluation metrics for retrieval performance, i.e., recall@2 (R@2) and recall@5 (R@5); two evaluation metrics for final QA performance, i.e., exact match (EM) score and F1 score; and finally, several widely used retrieval methods: BM25 (Robertson & Walker, 1994), Contriever (Izacard et al., 2022), GTR (Ni et al., 2022), ColBERTv2 (Santhanam et al., 2022), RAPTOR (Sarthi et al., 2024), Proposition (Chen et al., 2024b), GraphRAG (MS) (Edge et al., 2024), LightRAG (Guo et al., 2025), HippoRAG (Jimenez Gutierrez et al., 2024), SubgraphRAG (Li et al., 2025a), G-retriever (He et al., 2024), and GFM-RAG (Luo et al., 2025).

Regarding LP-RAG, we again employ the same setup from Section 4.1 and use `GPT-4.o-mini`.

Table 3: Retrieval performance comparison. LP-RAG outperforms GFM-RAG in three scenarios, and is second best in all scenarios.

| Method | HotpotQA | | MuSiQue | | 2Wiki | |
|---|---|---|---|---|---|---|
| | R@2 | R@5 | R@2 | R@5 | R@2 | R@5 |
| BM25 | 55.4 | 72.2 | 32.3 | 41.2 | 51.8 | 61.9 |
| Contriever | 57.2 | 75.5 | 34.8 | 46.6 | 56.0 | 57.5 |
| GTR | 59.4 | 73.3 | 37.4 | 47.9 | 60.2 | 67.0 |
| ColBERTv2 | 64.7 | 79.3 | 37.9 | 49.2 | 59.2 | 68.2 |
| RAPTOR | 58.1 | 75.9 | 35.7 | 49.3 | 56.9 | 63.0 |
| Proposition | 58.7 | 71.1 | 37.6 | 49.3 | 56.4 | 63.4 |
| GraphRAG (MS) | 58.3 | 76.6 | 35.4 | 49.3 | 61.6 | 77.3 |
| LightRAG | 38.8 | 54.7 | 31.4 | 43.9 | 51.0 | 67.9 |
| HippoRAG (Contriever) | 59.0 | 76.2 | 39.1 | 52.1 | 71.5 | 89.5 |
| HippoRAG (ColBERTv2) | 60.5 | 77.9 | 40.2 | 53.2 | 73.5 | 90.4 |
| SubgraphRAG | 61.5 | 73.0 | 36.9 | 50.2 | 63.3 | 71.7 |
| G-retriever | 53.3 | 65.5 | 38.8 | 45.1 | 60.8 | 67.5 |
| GFM-RAG | **78.3** | 87.1 | 49.1 | 58.2 | **90.8** | **95.6** |
| LP-RAG | 77.9 | **89.6** | **53.6** | **61.8** | 89.9 | 93.2 |

**Results: retrieval performance.** As shown in Table 3, LP-RAG achieves the best performance on three scenarios, outperforming the SOTA GFM-RAG by 4.5% in R@2 and 3.6% in R@5 on MuSiQue. The results demonstrate the effectiveness of LP-RAG in multi-hop retrieval. This results

Table 4: Results: Q&A performance. LP-RAG is the best method in both metrics for all datasets.

| Method | HotpotQA | | MuSiQue | | 2Wiki | |
|---|---|---|---|---|---|---|
| | EM | F1 | EM | F1 | EM | F1 |
| None | 30.4 | 42.8 | 12.5 | 24.1 | 31.0 | 39.0 |
| ColBERTv2 | 43.4 | 57.7 | 15.5 | 26.4 | 33.4 | 43.3 |
| GraphRAG (MS) | 35.3 | 54.6 | 13.6 | 24.9 | 28.3 | 38.4 |
| LightRAG | 36.8 | 48.3 | 18.1 | 27.5 | 45.1 | 49.5 |
| HippoRAG (ColBERTv2) | 41.8 | 55.0 | 19.2 | 29.8 | 46.6 | 59.5 |
| GFM–RAG | 51.6 | 66.9 | 30.2 | 40.4 | 69.8 | 77.7 |
| LP–RAG | **53.2** | **69.1** | **32.6** | **43.9** | **72.6** | **78.3** |

indicates that the LP-RAG can effectively conduct multi-hop reasoning. Compared to the other baselines, LP-RAG significantly outperforms them by a large margin. In particular, against HippoRAG (ColBERTv2), LP-RAG has a 17.4% gain in R@2 for the HotpotQA dataset.

**Results: Q&A performance.** Table 4 depicts the results regarding Q&A performance. Notably, LP-RAG achieves state-of-the-art performance against all baselines. In particular, LP-RAG outperforms GFM-RAG by 1.6%, 2.4%, 2.8% in EM on HotpotQA, MuSiQue, and 2Wiki, respectively.

## 4.4 EFFICIENCY ANALYSIS

We now compare the computational performance of LP-RAG with the main graph-based methods in Section 4.1. Our comparison covers indexing time, storage usage, average query latency, and average number of retrieved tokens. Importantly, all experiments were conducted under the same hardware and setup to ensure a fair comparison. We report the results in Table 5.

Overall, our method demonstrates equal or superior performance across all five datasets. In terms of query time, it is the fastest method on 4 out of 5 datasets, indicating that the overhead introduced by the GNN-based predictor is offset by the significantly lower number of retrieved tokens. Also, the indexing time is comparable with that of NodeRAG. These results underscore the efficiency of our framework when compared with existing graph-based approaches. For completeness, we also report the computational performance of LP-RAG on large datasets (Arena Benchmarks) in Section D.1.

Table 5: LP-RAG performance. Corpus size in total tokens, indexing time in minutes, storage usage in MegaBytes (MB), query time in seconds (s). Light refers to LightRAG and Node refers to NodeRAG

| Dataset | Size | Indexing time (min) | | | Storage usage (MB) | | | Avg query time (s) | | | Avg retrieved tokens | | |
|---|---|---|---|---|---|---|---|---|---|---|---|---|---|
| | | Light | Node | LP-RAG | Light | Node | LP-RAG | Light | Node | LP-RAG | Light | Node | LP-RAG |
| HotpotQA-S | 62K | 7.9 | 5.1 | 4.7 | 21.2 | 10.7 | 10.3 | 5.8 | 4.2 | 3.6 | 5462 | 4019 | 3208 |
| MuSiQue-S | 58K | 5.6 | 5.2 | 5.8 | 21.6 | 9.8 | 10.1 | 6.2 | 5.6 | 5.1 | 5312 | 5269 | 3849 |
| MultiHop-S | 49K | 5.9 | 5.3 | 4.8 | 19.7 | 10.4 | 9.7 | 7.4 | 5.9 | 5.1 | 6752 | 6032 | 3816 |
| RAG-QA-S | 54K | 6.1 | 5.8 | 6.2 | 26.8 | 14.0 | 12.1 | 5.6 | 4.8 | 4.2 | 6325 | 3370 | 2279 |
| ICLR-S | 67K | 10.3 | 8.6 | 7.9 | 38.7 | 26.9 | 15.7 | 7.9 | 5.4 | 5.7 | 6946 | 6214 | 2428 |

## 4.5 ABLATION STUDY

We now conduct an ablation study using the same five datasets and evaluation metrics as in Table 1. We begin by evaluating the contribution of the GNN-based retrieval module, then examine the effect of incorporating synthetic queries, and finally assess how different negative sampling strategies influence LP-RAG's performance.

**Impact of GNN architecture.** We consider four popular GNNs for link prediction: VGAE (Kipf & Welling, 2016), PEG (Wang et al., 2022), SEAL (Zhang & Chen, 2018), and NCC (Wang et al., 2024). For completeness, we also include an MLP. The results are given in Table 6.

Overall, the results confirm that a robust GNN with stronger negative link discrimination is essential for producing higher accuracy results. In contrast, an MLP, which lacks the capacity to model relationships between entities, tends to underperform even compared to NaiveRAG. Another notable

Table 6: Ablation study concerning the performance of different GNNs and the importance of synthetic queries. Here, ∗ denotes models trained using only chunk nodes (no synthetic queries). Overall, NCN is the best performing model, and using synthetic queries is crucial.

| GNN | HotpotQA-S | | | MuSiQue-S | | | MultiHop-S | | | RAG-QA-S | | | ICLR-S | | |
|---|---|---|---|---|---|---|---|---|---|---|---|---|---|---|---|
| | Acc.↑ | F1.↑ | Tok↓ | Acc.↑ | F1.↑ | Tok↓ | Acc.↑ | F1.↑ | Tok↓ | Acc.↑ | F1.↑ | Tok↓ | Acc.↑ | F1.↑ | Tok↓ |
| SEAL | 89.28 | 85.12 | 3.2k | 89.41 | 84.05 | 3.5k | 89.03 | 87.62 | 3.7k | 87.73 | 83.13 | 2.2k | 89.14 | 88.23 | 2.5k |
| PEG | 87.01 | 86.11 | 3.3k | 85.22 | 83.16 | 3.5k | 87.64 | 85.63 | 3.8k | 83.14 | 82.63 | 2.2k | 84.37 | 82.09 | 2.6k |
| VGAE | 82.03 | 80.43 | 3.2k | 84.63 | 80.50 | 3.7k | 84.07 | 81.64 | 3.7k | 83.29 | 80.19 | 2.4k | 86.21 | 79.53 | 2.6k |
| NCN | 89.32 | 86.29 | 3.2k | 89.67 | 85.15 | 3.8k | 89.12 | 86.35 | 3.8k | 87.89 | 85.26 | 2.2k | 91.16 | 89.51 | 2.4k |
| MLP | 32.50 | 12.25 | 5.3k | 43.91 | 22.50 | 5.6k | 29.80 | 16.24 | 5.7k | 42.58 | 16.80 | 5.7k | 48.90 | 23.78 | 5.3k |
| VGAE* | 60.50 | 42.82 | 3.9k | 50.92 | 41.63 | 3.9k | 40.06 | 22.16 | 4.1k | 60.15 | 37.16 | 2.8k | 60.15 | 48.60 | 2.8k |
| NCN* | 69.15 | 42.80 | 3.9k | 68.64 | 55.20 | 4.0k | 52.15 | 48.05 | 3.8k | 67.15 | 45.18 | 2.4k | 72.60 | 48.95 | 2.9k |

result is that VGAE achieves a result comparable to the state-of-the-art NodeRAG, even though VGAE is a much less simpler model than NCN. This suggests that even simpler GNNs can achieve high-quality chunk retrieval when trained using our approach.

**Impact of synthetic queries.** Our second experiment (also reported in Table 6) evaluates the impact of synthetic queries on retrieval quality. Here, the GNNs architectures were trained on graphs containing only interconnected chunks, without synthetic queries. In this setting, even the strongest GNN (i.e., NCN) performs similarly to NaiveRAG, confirming the importance of augmenting graph information with synthetic queries in our framework.

**Impact of negative sampling.** Recent works (Aiyappa et al., 2025) have discussed issues related uniform negative sampling in link prediction. To assess the impact of negative sampling, we compare LP-RAG using uniform vs. degree-based negative sampling (Aiyappa et al., 2025), under the setup of EXPERIMENT I. As we can observe, modern negative-sampling strategies improve the performance of LP-RAG in some cases. However, despite these improvements, their overall impact is considerably smaller than the effect of the GNN architecture. We hypothesize that this is likely due to the approximately regular structure of our query–chunk graph, which results from the use of a $k$-NN–based construction.

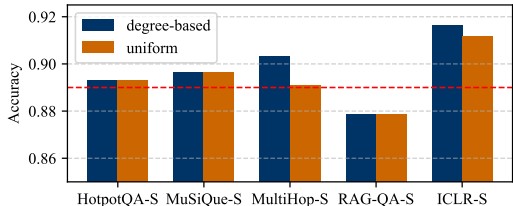

Figure 5: Impact of negative sampling on Q&A accuracy: Uniform vs. degree-based sampling.

**Additional results.** For completeness, in Appendix D, we report further experiments regarding i) the generalization of LP-RAG to unseen documents and cross-domain transferability; ii) the effect of our chunking scheme on other RAG methods; and iii) computational performance on large-scale data.

## 5 CONCLUSIONS

In this paper, we propose a novel graph-based RAG framework based on link prediction. Crucially, our method relies on synthetic queries to obtain supervision signals for the link prediction GNN. Intuitively, these queries enrich the graph with fine-grained semantic cues, guiding retrieval toward passages that are not only structurally connected but also semantically relevant. Retrieval tasks are formulated as inductive link prediction over a chunk–query graph. Through extensive experiments on multiple datasets, we demonstrate that LP-RAG not only achieves a significant improvement in accuracy and F1-score compared to multiple baselines but also operates with little to moderate resources. Overall, LP-RAG demonstrates that integrating synthetic queries into RAG pipelines can significantly boost the quality of information retrieval and performance on question-answering tasks.

**Limitations.** As most GraphRAG methods, scalability is a major issue. As the size of the corpus grows, the number of nodes and edges can quickly reach millions or bilions, leading to prohibitive storage and retrieval costs. Running graph neural networks on such large graphs is often computationally expensive and infeasible in real-time settings. In this context, adopting simple heuristics for link prediction represents a promising alternative, given their strong predictive performance.

ETHICS STATEMENT

We do not anticipate any immediate societal or ethical concerns arising from this work. In addition, we acknowledge the use of LLMs to assist with grammar and style checks in parts of the manuscript, primarily in the introduction.

REPRODUCIBILITY STATEMENT

To ensure the reproducibility of our work, we will release the code publicly after the review process is complete, including data splits and all prompts used. Furthermore, we provide implementation details in Appendix B.

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

## A   LP-RAG PIPELINE EXAMPLE

In this section, we attempt to show a simple example of chunks returned by LP-RAG and the LLM response for a given query. The pipeline followed in this experiment was the same as that reported in Table 1. In red we highlight the main chunks used by LLM to generate the response.

---

**Box A.1: Real query from MuSiQue**

**Query**:
When was the publisher of Crux launched?

**LP-RAG retrieved chunks with probabilities**:
"Crux Ansata is a book subtitled 'An Indictment of the Roman Catholic Church' by H. G. Wells.", 0.94
"Crux Ansata is a wartime book.", 0.87
"Crux Ansata has 96 pages.", 0.92
"Crux Ansata was first published in 1943.", 0.95
"Crux Ansata was published by Penguin Books.", 0.91
"Crux Ansata was published in Hammonsworth, Great Britain.", 0.88
"Crux Ansata is classified as Penguin Special No. 129.", 0.90
"The U.S. edition of Crux Ansata was copyrighted.", 0.89
"The U.S. edition of Crux Ansata was published in 1944.", 0.93
"The U.S. edition of Crux Ansata was published by Agora Publishing Company.", 0.86
"The U.S. edition of Crux Ansata was published in New York.", 0.92
"The U.S. edition of Crux Ansata includes a portrait frontispiece.", 0.90
"The U.S. edition of Crux Ansata includes an appendix.", 0.87
"The appendix of the U.S. edition of Crux Ansata contains an interview with H. G. Wells.", 0.94
"The interview with H. G. Wells in the U.S. edition of Crux Ansata was recorded by John Rowland.", 0.91
"The U.S. edition of Crux Ansata has 144 pages.", 0.88
"The U.S. edition of Crux Ansata went into a third printing.", 0.93
"The third printing of the U.S. edition of Crux Ansata occurred in August 1946.", 0.89
"CrossGen was an American comic book publisher.", 0.95
"CrossGen was an entertainment company.", 0.88
"CrossGen operated from 1998 to 2004.", 0.95
"The Walt Disney Company acquired CrossGen's assets in 2004.", 0.94
"CrossGen's assets were designated to Disney Publishing Worldwide.", 0.90
"Disney re-established the CrossGen brand in July 2010.", 0.87
"Disney re-established the CrossGen brand through Marvel Comics.", 0.91
"Marvel Comics announced plans to revive CrossGen titles.", 0.89
"Crux is an American comic book.", 0.86
"Crux was published by CrossGen Entertainment.", 0.98
"Crux was published from May 2001 to February 2004.", 0.88
"Crux was cancelled in 2004.", 0.92
"Crux was cancelled due to bankruptcy.", 0.87
"Crux was one of the later titles of Crossgen.", 0.91
"Crux came first in a second wave of Crossgen titles.", 0.89
"The second wave of Crossgen titles included Sojourn.", 0.85
"The second wave of Crossgen titles included Brath.", 0.88
"Crux detailed the exploits of six Atlanteans.", 0.93
"The six Atlanteans were put into stasis.", 0.90
"The six Atlanteans were awoken 100,000 years later.", 0.94

**LLM response**:
The publisher of Crux was launched in 1998 by CrossGen that operated from 1998 to 2004.

---

**Why does it work?** During the chunk-node graph generation phase, LP-RAG interconnected most of the returned chunks in this example. In particular, the chunks ["CrossGen operated from 1998 to 2004.", "Crux was published by CrossGen Entertainment.", "CrossGen was an American comic book publisher.", "Crux is an American comic book."] were interconnected. During the synthetic query generation phase for the chunks, our model had the following queries added to the LP-RAG training graph: ["Which American comic book publisher, active between 1998 and 2004, published Crux?", "Which company published American comic books between 1998 and 2004, including Crux?", "What is Crux and who published it?"]. This combination of synthetic queries and interconnected chunks during the training stage provided LP-RAG with a high power of interpretation of the query in question in the example, thus returning the correct answer.

## B  EXPERIMENTAL DETAILS

### B.1  DATASETS AND BASELINES

**Datasets.** We consider seven benchmarks:

- *HotpotQA* (Yang et al., 2018) is a multi-hop question-answering dataset where each question requires combining information from multiple documents to find the correct answer.
- *MuSiQue* (Trivedi et al., 2022) is also a multi-hop question-answering dataset that challenges models to combine information across multiple documents in a structured, step-by-step manner.
- *MultiHop-RAG* (Tang & Yang, 2024) is a multi-hop question-answering dataset that includes four distinct question types: comparison query, null query, inference query, and temporal query.
- *Arena benchmark* (Han et al., 2024b) builds on Long-form RobustQA (LFRQA), a dataset of 26K queries across 7 domains. Here, we consider the Writing, Tech, Science, and Recreation domains.
- *RAG-QA* (Han et al., 2024b) is a shortened version of the Arena Benchmark, where we selected samples from the 7 domains.
- *ICLR Papers* (Feng et al., 2025) is formed by abstracts and review decisions from paper submissions to the ICLR conferences between 2021 and 2023. We use an LLM to generate questions and answers to tailor it to the Q&A task.
- *2WikiMultiHopQA (2Wiki)* (Ho et al., 2020) is a multi-hop QA dataset that requires reasoning over multiple Wikipedia articles to answer questions. The dataset consists of 192k questions, which are designed to be answerable using information from 2 or 4 articles.

**Baselines.** In the following, we provide a brief overview of the baselines considered in the paper.

- *NaiveRAG* (Lewis et al., 2020) represents a simple baseline for RAG systems. It first divides documents into several text chunks and encode them into a vector space utilizing text embeddings. Then retrieve related text chunks based on similarity of query representations.
- *HyDe* (Gao et al., 2023a) improves upon traditional RAG systems. In particular, HyDe uses an LLM to generate "synthetic answers" from the input query. These "synthetic answers" are transformed into embeddings and used to search for the most similar chunks.
- *GraphRAG* (Edge et al., 2024) segments the input document into chunks and extracts the entities and their relationships to create a graph. This graph is then divided into multiple communities at different levels. At query time, GraphRAG identifies the relevant entities from the question and synthesizes answers by referencing these corresponding community summaries.
- *LightRAG* (Guo et al., 2025) is an improved approach based on GraphRAG, designed to minimize computational overhead while enhancing the comprehensiveness of retrieved information through dual-level retrieval. This leads to more efficient retrieval and a better balance between effectiveness and speed compared to GraphRAG.
- *NodeRAG* (Xu et al., 2025) is a graph-centric approach that leverages heterogeneous graphs.
- *BM25* (Robertson & Walker, 1994) is a classic information retrieval method based on the probabilistic model that ranks a set of documents based on the query terms frequency appearing in each document.

- *Contriever* (Izacard et al., 2022) trains a dense retriever with contrastive learning on a large-scale corpus to retrieve relevant documents for a given query.
- *GTR* (Ni et al., 2022) develops a scale-up T5-based dense retriever that could generalize across different datasets and domains.
- *ColBERTv2* (Santhanam et al., 2022) is a state-of-the-art dense retriever that couples an aggressive residual compression mechanism with a denoised supervision strategy to simultaneously improve the retrieval quality.
- *RAPTOR* (Sarthi et al., 2024) is an LLM-augmented retriever that recursively embeds, clusters, and summarizes chunks of text, constructing a tree with differing levels of summarization to enable accurate retrieval.
- *Proposition* (Chen et al., 2024b) enhances the performance of dense retrievers by leveraging LLMs to generate a natural language proposition that captures the essential information of the document.
- *HippoRAG* (Jimenez Gutierrez et al., 2024) is a state-of-the-art, training-free graph-enhanced retriever that uses the Personalized PageRank algorithm to assess entity relevance to a query and performs multihop retrieval on a document-based knowledge graph. It can be directly applied to various datasets.
- *GFM-RAG* (Luo et al., 2025) uses a Graph Foundation Model (GFM) to reason over a knowledge-graph index built from documents. The model is pretrained at scale (60 KGs, 14M triples + 700k documents).
- *SubgraphRAG* (Li et al., 2025a) retrieves relevant subgraphs from a knowledge graph (KG) using a lightweight multilayer perceptron and a parallel triple-scoring mechanism, while encoding directional structural distances to improve retrieval effectiveness. Then, those subgraphs (triples) are fed into an LLM, which reasons over them to generate grounded and explainable answers.
- *G-retriever* (He et al., 2024) works by: (1) indexing graph nodes and edges using embeddings; (2) retrieving the most relevant elements based on similarity when given a query; (3) constructing a relevant subgraph by formulating it as a Steiner Tree problem to avoid context explosion; and (4) using an LLM to generate an answer with soft prompt tuning based on that subgraph.

## B.2 DETAILS ON EXPERIMENT I

In this section, we provide more details about the evaluation setup for the experiments reported in Tables [1, and 6], and about the hyperparameters used in all GNNs.

LP-RAG was implemented using PyTorch, with all experiments conducted on a machine with 32GB RAM, Intel Core I7-13700 30mb 2.5ghz, and a single NVIDIA RTX 2060 super with 8GB. In addition, we used `LLaMA 3.3-70B` with temperature 0 and API calls from groq API.

For MuSiQue, we use the "ans" version. For the other datasets, we use the version available in their respective repositories. We randomly sampled 20 documents from each dataset, with the exception of the ICLR papers, where we sampled 50 documents since it is much smaller.

Table 7: Hyperparameters used for the link prediction methods.

| GNN | LR | WD | layers | epochs | h-dim |
|------|------|------|--------|--------|-------|
| VGAE | $10^{-2}$ | $10^{-4}$ | 2 | 400 | 32 |
| PEG | $10^{-2}$ | $10^{-4}$ | 2 | 400 | 32 |
| NCN | 0.0043 | $10^{-8}$ | 1 | 100 | 256 |
| SEAL | $10^{-4}$ | $10^{-8}$ | 3 | 50 | 32 |
| MLP* | $10^{-3}$ | $5 * 10^{-4}$ | 2 | 1000 | 1000 |

Regarding the configuration of each GNN, Table 7 shows the used hyperparameters. To train the GNNs, we employ the Adam optimizer (Adam et al., 2014). We use a uniform random negative sampling strategy. We considered chunks with a confidence level greater than 50% ($\tau = 0.5$) assigned by the GNN to generate the response.

### B.3  DETAILS ON EXPERIMENT II

For Experiment II, we use NCN with the same hyperparameter choice reported in Table 7. We also use the same negative sampling, Adam optimizer, and chunk selection criteria from EXPERIMENT I. Also, we use `gpt-4o-mini` with temperature 0 as LLM with calls to the OpenAI API.

For Arena-Writing, we consider 11,250 documents for chunking, and 2,693 questions. For Arena-Science, we consider 5,283 documents and 1,403 questions. For Arena-Tech, we consider 8,438 documents and 2,063 questions. Finally, for Arena-Recreation, we consider 4,341 documents and 2,089 questions. The baseline results reported in Table 2 were collected from (Xu et al., 2025).

### B.4  DETAILS ON EXPERIMENT III

For EXPERIMENT III, we also use NCN as link predictor with uniform negative sampling. Table 8 shows the hyperparameters for each dataset. We use `gpt-4o-mini` with temperature 0 as LLM.

Table 8: Hyperparameters for NCN used in Experiment III.

| dataset | LR | WD | layers | epochs | h-dim |
|---------|------|--------|--------|--------|-------|
| MuSiQue | 0.0043 | $10^{-8}$ | 1 | 1000 | 256 |
| HotpotQA | 0.0043 | $10^{-8}$ | 3 | 1000 | 256 |
| 2wiki | 0.0085 | $10^{-8}$ | 3 | 1000 | 32 |

MuSiQue, HotpotQA, and 2wiki have 6119, 9221, 11656 documents, respectively. Each dataset contains 1000 documents for test. To train LP-RAG, we use the same test split reported from Luo et al. (2025). The baseline results reported in Table 3 and Table 4 were collected from (Luo et al., 2025).

## C  USED PROMPTS

In this section, we provide all the prompts used during the LP-RAG experiments. Note that, overall, we adopted a few-shot example instruction prompting strategy. This allowed us to have more control over the LLM output, making the task of retrieving information easier via Regex. For example, by specifying in the prompt that we want the output of the chunk extraction step to be [chunk], we can force LLM to add the characters [ and ] to each chunk, which allows for very easy string regex searching. One drawback of using this technique is that it significantly increases LLM's input token consumption, which can result in higher additional costs for users of paid APIs, such as openAI. However, openAI itself is already addressing this issue by implementing the cached-prompt strategy.

**Box C.1: Chunk extraction prompt**

**—Role: System—**

You are required to act as an AI annotator and extract the Chunks embedded in the sentences of the provided paragraph text. Below, you will be given a paragraph from a text. You need to break it down sentence by sentence and extract the Chunks embedded in each sentence. The extracted Chunks can be an idea, argument, or fact. Each sentence may contain one or more Chunks to be extracted. The extracted Chunks should be as granular as possible to ensure they cannot be further broken down.

When extracting Chunks from a sentence, pay attention to the context within the paragraph. Replace pronouns with the nouns they represent and complete any omitted sentence components to ensure the independence of the Chunks is not compromised. This means that each extracted Chunk should not contain pronouns whose referents cannot be found within that Chunk.

Below is an example interaction that can serve as a reference for format and method of extracting Chunks:

System's Input:

**[The Start of Paragraph text]**
Pakistan Super League (Urdu: PSL) is a Twenty20 cricket league, founded in Lahore on 9 September 2015 with five teams and now comprises six teams. Instead of operating as an association of independently owned teams, the league is a single entity in which each franchise is owned and controlled by investors.
**[The End of Paragraph text]**

Your Answer:

**[Sentence 1]**
Pakistan Super League (Urdu: PSL) is a Twenty20 cricket league, founded in Lahore on 9 September 2015 with five teams and now comprises six teams.

**[Extracted Chunks in Sentence 1]**
[Pakistan Super League (PSL) is a Twenty20 cricket league.]
[Pakistan Super League (PSL) was founded in Lahore on 9 September 2015.]
[Pakistan Super League (PSL) was founded with five teams.]
[Pakistan Super League (PSL) now comprises six teams.]

**[Sentence 2]**
Instead of operating as an association of independently owned teams, the league is a single entity in which each franchise is owned and controlled by investors.

**[Extracted Chunks in Sentence 2]**
[Pakistan Super League (PSL) does not operate as an association of independently owned teams.]
[Pakistan Super League (PSL) operates as a single entity.]
[Each franchise in the Pakistan Super League (PSL) is owned by investors.]
[Each franchise in the Pakistan Super League (PSL) is controlled by investors.]

**—Role: User—**
**[The Start of Paragraph text]**
{Your Text Here}
**[The End of Paragraph text]**

**Box C.2: Query response prompt**

**—Role: System—**
You are a through assistant responding to questions based on retrieved information.
Below, you'll find a bag of chunks and a query. Your job is to answer the query using only the information contained in the chunks. If the answer isn't in the chunks, simply answer: "I don't know."

**—Role: User—**
**[Bag of chunks]**
{Retrieved chunks}

**[Query]**
{Your query}

---

**Box C.3: Synthetic query generation prompt**

**—Role: System—**
You are required to act as an AI annotator and generate questions in the provided paragraph chunks. Below, you will be given a bage of chunks from a paragraph. You need to generate two questions about it for each chunk. The generated two questions need to be like a human made question and each chunk may contain two questions. The question generated should be as similar as possible to a human question.
When generating the two questions from a chunk, pay attention to the context within the chunk.

Below is an example interaction that can serve as a reference for format and method of generate questions:

System's Input:

**[The Start bag of chunks]**
**[Chunk 1]**
We propose a model-based reinforcement learning (MBRL) approach for continuous action spaces.
**[Chunk 2]**
Bthardamz is only accessible during the quest The Only Cure.
**[The End bag of chunks]**

Your Answer:

**[Questions Chunk 1]**
[What type of reinforcement learning approach was proposed?]
[What type of action spaces does the proposed approach target?]
**[Questions Chunk 2]**
[When can Bthardamz be accessed?]
[What quest must be completed to access Bthardamz?]

**—Role: User—**
**[The Start bag of chunks]**
{Your chunks here}
**[The End bag of chunks]**

# D  ADDITIONAL EXPERIMENTS

## D.1  EFFICIENCY

In this section, we compare the computational efficiency of LP-RAG against NodeRAG (Xu et al., 2025) on the Arena benchmarks. For indexing time, we consider the time to extract the chunks, plus the time to generate the synthetic queries, plus the time to train the GNN. Corpus size is in total tokens; storage usage comprises the size of the final generated graph plus the parameters of the GNN. Avg Query time is the average response time of the entire pipeline (in seconds).

Table 9: Performance metrics for LP-RAG method, including Indexing Time, Storage Usage, Avg Query Time, and Retrieval Tokens across Arena datasets.

| Dataset | Corpus size | Index time | | Storage usage | | Avg Query time | | Avg retrieved tokens | |
|---|---|---|---|---|---|---|---|---|---|
| | | NodeRAG | LP-RAG | NodeRAG | LP-RAG | NodeRAG | LP-RAG | NodeRAG | LP-RAG |
| A. Writing | 1.82M | 13min | 183min | 157MB | 175MB | 5.40s | 6.32s | 3373 | 4219 |
| A. Tech | 1.72M | 14min | 165min | 139MB | 168MB | 6.74s | 7.98s | 3821 | 4863 |
| A. Science | 1.43M | 17min | 161min | 111MB | 157MB | 8.85s | 8.69s | 4284 | 4188 |
| A. Recreation | 0.93M | 10min | 152min | 80MB | 143MB | 6.90s | 7.64s | 3448 | 4103 |

Table 9 presents the system performance of the NodeRAG method and our proposed approach. Compared to NodeRAG, LP-RAG achieves comparable performance in terms of query time, storage usage, and retrieved tokens.

## D.2  IMPACT OF CHUNKING STRATEGY

The chunking strategy can be considered a data preprocessing step and therefore can impact the method's performance. In this experiment, we analyzed whether our chunk strategy could benefit other baselines. We used the chunks extracted from the samples used in EXPERIMENT I to feed all the baselines and we report the macro accuracy found.

Table 10: Macro accuracy using our chunk strategy as data preprocessing for the baselines. LP-RAG results are from Table 1.

| Method | HotpotQA-S | MuSiQue-S | MultiHop-S | RAG-QA-S | ICLR-S |
|---|---|---|---|---|---|
| NaiveRAG | 18.33 | 15.56 | 16.67 | 12.78 | 11.11 |
| HyDe | 25.46 | 21.19 | 36.50 | 32.67 | 37.69 |
| LightRAG | 71.25 | 60.62 | 64.25 | 61.20 | 66.62 |
| NodeRAG | 83.31 | 62.29 | 70.64 | 67.05 | 74.43 |
| **LP-RAG** | **89.32** | **89.67** | **89.12** | **87.89** | **91.16** |

As we can see in Table 10, our chunk strategy does not show a significant gain for the baselines; in fact, it's the opposite. All baselines showed lower accuracy than those reported in Table 1. The most impacted baseline was NaiveRAG, with all results falling below 20%. We believe that our short chunks with atomic information are too small for similarity-based methods to correctly connect them with a query. Larger chunks provide more information, and traditional RAG models can take advantage of this to generate a more accurate response, which is not the case when they use our chunking strategy.

## D.3  GENERALIZATION TO UNSEEN DOCUMENTS (SMALL-DATA REGIME)

In this section, we evaluate the generalizability of LP-RAG by conducting domain-specific generalization on data not seen during training. For each domain-specific LP-RAG acquired during EXPERIMENT I, we select 50 completely new samples. For this new data, we only construct the chunk graph, without fine-tuning the model. We also re-evaluated the baselines with the same 50 samples. In the end, we calculated the macro accuracy.

As can be seen in Table 11, LP-RAG is able to generalize to new documents not seen during training. In fact, macro accuracy has a significant decay compared to the results in Table 1, but our method is still superior in all datasets, especially MuSiQue, which had a 16% gain compared to NodeRAG.

Table 11: Generalization to unseen documents.

| Method | HotpotQA-S | MuSiQue-S | MultiHop-S | RAG-QA-S | ICLR-S |
|---|---|---|---|---|---|
| NaiveRAG | 52.00 | 42.00 | 38.00 | 54.00 | 64.00 |
| HyDe | 64.00 | 68.00 | 66.00 | 62.00 | 70.00 |
| LightRAG | 74.00 | 62.00 | 68.00 | 66.00 | 72.00 |
| NodeRAG | 84.00 | 66.00 | 74.00 | 68.00 | 72.00 |
| **LP-RAG** | **86.00** | **82.00** | **84.00** | **82.00** | **84.00** |

### D.3.1 CROSS-DOMAIN TRANSFERABILITY

To demonstrate the transferability of LP-RAG, we consider the retrieval performance (R@2 and R@5)of LP-RAG on two datasets following the setup of Experiment III. In particular, We now take the LP-RAG model trained on the MuSiQue dataset and evaluate it on HotpotQA and 2Wiki.

Table 12: Retrieval performance comparison of LP-RAG trained only using the MuSiQue dataset.

| Method | HotpotQA | | 2Wiki | |
|---|---|---|---|---|
| | R@2 | R@5 | R@2 | R@5 |
| BM25 | 55.4 | 72.2 | 51.8 | 61.9 |
| Contriever | 57.2 | 75.5 | 56.0 | 57.5 |
| GTR | 59.4 | 73.3 | 60.2 | 67.0 |
| ColBERTv2 | 64.7 | 79.3 | 59.2 | 68.2 |
| RAPTOR | 58.1 | 75.9 | 56.9 | 63.0 |
| Proposition | 58.7 | 71.1 | 56.4 | 63.4 |
| GraphRAG (MS) | 58.3 | 76.6 | 61.6 | 77.3 |
| LightRAG | 38.8 | 54.7 | 51.0 | 67.9 |
| HippoRAG (Contriever) | 59.0 | 76.2 | 71.5 | 89.5 |
| HippoRAG (ColBERTv2) | 60.5 | 77.9 | 73.5 | 90.4 |
| SubgraphRAG | 61.5 | 73.0 | 63.3 | 71.7 |
| G-retriever | 53.3 | 65.5 | 60.8 | 67.5 |
| GFM-RAG | **68.8** | **81.8** | **84.4** | 89.6 |
| **LP-RAG** | 65.4 | 78.3 | 75.9 | **91.6** |

As shown in Table 12, LP-RAG is the best model in 1/4 cases and the second-best in the remaining ones — surpassing models like HippoRAG by a large margin. It is important to note that GFM-RAG is pre-trained using a larger number of documents.

