# OpenReview forum: "LP-RAG: A Link Prediction-Based Framework for Retrieval-Augmented Generation"
_ICLR.cc/2026/Conference — Submitted to ICLR 2026_

### Official Review · Reviewer_M6Nt · 2025-10-30

**Soundness:** 3
**Presentation:** 3
**Contribution:** 2
**Rating:** 4
**Confidence:** 4

**Summary:**

This paper introduces LP-RAG, a novel framework that reformulates the retrieval step in Retrieval-Augmented Generation as an inductive link prediction task on a "chunk–query" graph. The core idea involves using an LLM to segment documents into fine-grained semantic chunks and generate synthetic queries for each, constructing a graph where edges represent chunk-query associations. A link prediction model is trained to score these connections, and at inference, a user query is treated as a new node; its link probabilities to all chunks are used for retrieval. Experiments demonstrate that LP-RAG outperforms existing graph-based RAG methods on several benchmarks while significantly reducing token consumption.

**Strengths:**

​​1  Framing retrieval as a link prediction problem is innovative. It effectively leverages the rich methodology of graph learning, opening a new direction for RAG research.

2 The use of LLM-generated synthetic queries as weak labels provides valuable supervisory signals. The ablation studies convincingly demonstrate the importance of this design choice for performance gains.

​​3 The document chunking strategy via a prompted LLM is a principled alternative to sliding windows, inherently reducing chunk size and directly contributing to the framework's token efficiency.

**Weaknesses:**

​​1  The computational complexity and resource demands of the proposed method for large-scale corpora are a primary concern. While the method is evaluated on small-scale datasets, its practicality for real-world applications with millions of chunks is questionable. The analysis in Section 3.3 is insufficient to ensure scalability.

​​2  The framework's performance heavily depends on the quality and diversity of the LLM-generated queries. The current strategy to increase diversity lacks quantitative validation. Redundant or low-quality synthetic queries could weaken the supervisory signal and harm generalization. Furthermore, the increased complexity and potential instability introduced by relying on an LLM for both chunking and query generation warrant deeper investigation.

3  Although token cost is reported, the critical metric of end-to-end latency for a user query is absent. The inference step involves computing a node embedding and running a full GNN forward pass, which could introduce significant delay. This latency might offset the benefits gained from token savings, and its absence from the evaluation is a notable omission.

4  The use of a uniform random negative sampling strategy, as mentioned in Section 3.4, is simplistic and likely introduces false negatives. In a k-NN graph, high-degree chunks may be semantically related to many queries, and random sampling could incorrectly label potential positives as negatives. The lack of experimentation with more sophisticated strategies like hard-negative mining is a weakness, as it may limit learning signal quality and final task performance.

**Questions:**

​​1 To address concerns about the synthetic queries inadvertently overlapping with the test set distribution, it would be better to perform a similarity analysis between the synthetic and real test queries. If a high similarity is found, an ablation study removing the high-similarity synthetic queries and retraining the model would help verify if performance is inflated by potential leakage.

2 How about the performance on large-scale datasets and the relationship between graph size (number of nodes/edges), memory usage, and inference latency?

---

> ### Author Response · Authors · 2025-11-21
> **Response: Part 1/2**
>
> We appreciate your feedback. We note that we have updated our manuscript based on your suggestions. Changes (denoted in blue) include comparison against additional baselines on larger datasets, ablation studies (impact of negative sampling), assessment of generalizability/transferability to unseen documents, and further efficiency analyses.
>
> Below, we provide point-by-point responses to your comments and we will be happy to engage in further discussion during the rebuttal period.
>
> > "W1 The computational complexity and resource demands of the proposed method for large-scale corpora are a primary concern. While the method is evaluated on small-scale datasets, its practicality for real-world applications with millions of chunks is questionable. The analysis in Section 3.3 is insufficient to ensure scalability."
>
> Thank you for your comment regarding scalability. LP-RAG has competitive efficiency compared to SOTA graph-based and KG-based RAG methods. Our method also has strong cross-domain knowledge transfer capabilities, which reduces the computational cost of retraining and generating new synthetic queries for new datasets.
>
> To assess the efficiency and scalability of LP-RAG, we have run two additional sets of experiments. The first one compares against the best-performing graph-based competitors: LightRAG and NodeRAG, under the same hardware and setup to ensure a fair comparison. Our comparison covers indexing time (chunk generation + synthetic query generation + GNN training), storage usage (GNN parameters + size of resulting graph), average query latency, and average number of retrieved tokens. The results are below.
>
> |**Dataset**|**Size**|**Indexing time (min)**|**Storage (MB)**| **Query time (s)**|**Tokens**|
> |-|-|-|-|-|-|
> |||Light · Node · Ours|Light · Node · Ours|Light · Node · Ours|Light · Node · Ours|
> |HotpotQA|62K|7.9 · 5.1 · 4.7|21.2 · 10.7 · 10.3|5.8 · 4.2 · 3.6|5462 · 4019 · 3208|
> |MuSiQue| 58K|5.6 · 5.2 · 5.8|21.6 · 9.8 · 10.1|6.2 · 5.6 · 5.1 |5312 · 5269 · 3849|
> |MultiHop| 49K|5.9 · 5.3 · 4.8|19.7 · 10.4 · 9.7|7.4 · 5.9 · 5.1|6752 · 6032 · 3816|
> |RAG-QA|54K|6.1 · 5.8 · 6.2|26.8 · 14.0 · 12.1|5.6 · 4.8 · 4.2|6325 · 3370 · 2279|
> |ICLR|67K|10.3 · 8.6 · 7.9|38.7 · 26.9 · 15.7|7.9 · 5.4 · 5.7|6946 · 6214 · 2428|
>
> Overall, our method demonstrates equal or superior performance across all five datasets. In terms of query time, it is the fastest method on 4 out of 5 datasets, indicating that the overhead introduced by the GNN-based predictor is offset by the significantly lower number of retrieved tokens.
>
> For completeness, we have also considered larger datasets. In particular, we compare LP-RAG with NodeRAG using the same metrics on Arena Benchmarks (thousands of documents). The results are:
>
> |**Dataset**|**Size**|**Indexing time (min)**|**Storage (MB)**|**Query time (s)**|**Tokens**|
> |-|-|--|-|-|-|
> |||Node Ours|Node Ours|Node Ours|Node Ours|
> |A. Writing|1.82M|13 · 183|157 · 175|5.40 · 6.32|3373 · 4219|
> |A. Tech|1.72M|14 · 165|139 · 168|6.74 · 7.98|3821 · 4863|
> |A. Science|1.43M|17 · 161|111 · 157|8.85 · 8.69|4284 · 4188|
> |A. Writing|0.93M|10 · 152|80 · 143|6.90 · 7.64|3448 · 4103|
>
> While LP-RAG introduces additional overhead in indexing time, its average query time remains comparable to that of existing methods. Given the strong empirical gains, we believe these results demonstrate both the feasibility and the practical potential of our approach on large datasets.
>
> > "W2 The framework's performance heavily depends on the quality and diversity of the LLM-generated queries..."
>
> Thank you for highlighting the importance of query diversity and the potential risks associated with LLM-generated supervision. In Table 6 (Section 4.5) of the updated PDF, we analyze the impact of removing synthetic queries during LP-RAG training. Our findings indicate that the synthetic queries produced by the LLMs used in this work are of sufficiently high quality to meaningfully influence model performance.
>
> Moreover, recent works such as NodeRAG, GraphRAG, and LightRAG (which rely on LLMs to generate KGs), as well as GraphEVAL [1] (which uses LLMs to extract atomic viewpoints), further demonstrate that LLMs are capable of extracting rich information from text.
>
> > W3 Although token cost is reported, the critical metric of end-to-end latency for a user query is absent. The inference step involves computing a node embedding and running a full GNN forward pass, which could introduce significant delay. This latency might offset the benefits gained from token savings, and its absence from the evaluation is a notable omission.
>
> Thanks for you comment. We have reported latency time in our answer to "W1" above. As you correctly noted, the overhead introduced by the GNN-based predictor is compensated by the substantially lower number of retrieved tokens.
>
> **(to be continued)**

---

> ### Author Response · Authors · 2025-11-21
> **Response: Part 2/2**
>
> > "W4 The use of a uniform random negative sampling strategy, as mentioned in Section 3.4, is simplistic and likely introduces false negatives. In a k-NN graph, high-degree chunks may be semantically related to many queries, and random sampling could incorrectly label potential positives as negatives. The lack of experimentation with more sophisticated strategies like hard-negative mining is a weakness, as it may limit learning signal quality and final task performance."
>
> Thank you for your comment. We conducted an additional experiment to evaluate LP-RAG using the recently introduced degree-based negative sampling scheme of [2]. The results are presented in the table below.
>
> |**Neg sampling**|**HotpotQA**|**MuSiQue**|**MultiHop**|**RAG-QA**|**ICLR**|
> |-|-|-|-|-|-|
> |uniform|89.32|89.67|89.12|87.89|91.16|
> |degree-based|89.32|89.67|90.33|87.89|91.66|
>
> As we can observe, the sampling scheme has little impact on accuracy. We hypothesize that this is due to the approximately regular structure of our query-chunk graph, which results from the use of a k-NN-based construction. We have added these results in our updated PDF (Section 4.5).
>
> > Q1 "To address concerns about the synthetic queries inadvertently overlapping with the test set distribution, it would be better to perform a similarity analysis between the synthetic and real test queries. If a high similarity is found, an ablation study removing the high-similarity synthetic queries and retraining the model would help verify if performance is inflated by potential leakage."
>
> Thank you for your comment. While we agree that monitoring overlap between synthetic and real test queries is important, we emphasize that some degree of similarity is both intended and beneficial: the goal of synthetic query generation is to approximate the real query distribution, thereby providing the model with supervision signals that closely mirror actual information needs. We also highlight that our model is fully trained in a self-supervised way (with no information from actual queries).
>
> Importantly, we have run additional experiments to ensure the generalization capabilities of our method. In particular, we consider two tasks: generalization to unseen documents within the same domain (dataset); cross-domain transferability. In the former, we set apart a collection of unseen documents during training, and incorporate only their chunk-graph for testing --- we did not retrain the GNN nor generate synthetic queries. In the latter, we train our model using one dataset, and assess how it performs on other ones (of a different domain). In particular, we trained LP-RAG on MuSiQue and assessed it on Hotpot and 2Wiki.
>
> The results for generalization to unseen documents are below. As we can see, LP-RAG remains the best model, outperforming GraphRAG, NodeRAG, and LightRAG.
>
> |**Method**|**HotpotQA-S**|**MuSiQue-S**|**MultiHop-S**|**RAG-QA-S**|**ICLR-S**|
> |------------|--------------|-------------|--------------|------------|----------|
> |NaiveRAG|52.00|42.00|38.00|54.00|64.00|
> |HyDe|64.00|68.00|66.00|62.00|70.00|
> |LightRAG|74.00|62.00|68.00|66.00|72.00|
> |NodeRAG|84.00|66.00|74.00|68.00|72.00|
> |LP-RAG|**86.00**|**82.00**|**84.00**|**82.00**|**84.00**|
>
> The results for cross-domain transferability are given in the following. LP-RAG is the best model in 1/4 cases and the second-best in the remaining ones.
>
> |**Method**|**HotpotQA R@2**|**HotpotQA R@5**|**2Wiki R@2**|**2Wiki R@5**|
> |-|-|-|-|-|
> |BM25|55.4|72.2|51.8|61.9|
> |Contriever|57.2|75.5|56.0|57.5|
> |GTR|59.4|73.3|60.2|67.0|
> |ColBERTv2|64.7|79.3|59.2|68.2|
> |RAPTOR|58.1|75.9|56.9|63.0|
> |Proposition|58.7|71.1|56.4|63.4|
> |GraphRAG (MS)|58.3|76.6|61.6|77.3|
> |LightRAG|38.8|54.7|51.0|67.9|
> |HippoRAG (Contriever)|59.0|76.2|71.5|89.5|
> |HippoRAG (ColBERTv2)|60.5|77.9|73.5|90.4|
> |SubgraphRAG|61.5|73.0|63.3|71.7|
> |G-retriever|53.3|65.5|60.8|67.5|
> |GFM-RAG|**68.8**|**81.8**|**84.4**|89.6|
> |**LP-RAG**|65.4|78.3|75.9|**91.6**|
>
> Overall, these results demonstrate the strong transferability of LP-RAG: it can be applied directly to unseen datasets without any domain-specific fine-tuning, while still delivering strong results.
>
> > "Q2 How about the performance on large-scale datasets and the relationship between graph size (number of nodes/edges), memory usage, and inference latency?"
>
> We report memory usage and inference latency on Arena Benchmarks in our answer to "W1".
>
> References:
>
> _[1] GraphEval: A light weight graph-based llm framework for idea evaluation. ICLR, 2025_
>
> _[2] Implicit degree bias in the link prediction task. ICML, 2025._
>
> ---
>
> We hope your concerns have been satisfactorily addressed, and if so, would appreciate it if you could revisit your score to reflect the same.

---

### Official Review · Reviewer_9Z9c · 2025-10-30

**Soundness:** 3
**Presentation:** 3
**Contribution:** 3
**Rating:** 4
**Confidence:** 4

**Summary:**

LP-RAG reframes RAG retrieval as link prediction on a graph. Documents are split into chunks, chunks are linked by semantic similarity, and each chunk gets synthetic queries that connect to it, forming a chunk–query graph. A link-prediction model is trained to decide whether a query should connect to a chunk; at inference it predicts the best chunks to retrieve for generation, yielding higher accuracy and lower context cost than common GraphRAG baselines.

**Strengths:**

Strengths:

- Learning-based retrieval: Casts retrieval as link prediction, letting the retriever improve beyond static similarity.

- Encodes query intent: Synthetic queries write likely user phrasing directly into the index, boosting recall for varied asks.

- Token efficiency: Tends to surface shorter, high-signal chunks → smaller context windows.

**Weaknesses:**

Weaknesses

1. Missing key baselines (e.g., HippoRAG2, GFM-RAG).
   The evaluation omits strong contemporary graph-RAG methods, making it hard to gauge progress against the current state of the art.

2. Scalability challenges.
   The graph grows quickly with chunks and synthetic-query nodes/edges, and link-prediction models (e.g., GNN-based) can become memory- and time-intensive on large corpora.

3. Transferability.
   How transferable is the approach?: adapting to a new corpus typically requires regenerating synthetic queries and (re)training the link-prediction model rather than zero-shot transfer.

**Questions:**

**Transferability.**
   How transferable is the approach?: adapting to a new corpus typically requires regenerating synthetic queries and (re)training the link-prediction model rather than zero-shot transfer.

---

> ### Author Response · Authors · 2025-11-21
> **Response: Part 1/2**
>
> Thank you for your feedback. We have updated our PDF to address your concerns (changes are denoted in blue), including comparison with new baselines, efficiency analyses, and transferability. We reply to your comments/questions below.
>
> > "W1. Missing key baselines (e.g., HippoRAG2, GFM-RAG)..."
>
> Thanks for your comment. We have conducted additional experiments to compare our method with the baselines reported in [GFM, 1]. These experiments were carried out on MuSiQue, HotpotQA, and 2Wiki, following the evaluation protocol described in [GFM, 1]. To ensure a fair comparison, we used the same data splits, LLM configurations, and performance metrics as in those works.
>
> Below, we report performance regarding retrieval, Q&A, and computational performance in the single-step setting. Further implementation details are provided in Appendix B of the revised manuscript.
>
> **Retrieval performance**. Table below shows the recall@2 and recall@5 for LP-RAG and baselines. As we can observe, LP-RAG is the best model in 3/6 settings, and second best in the remaining ones.
>
> |**Method**|**HotpotQA R@2**|**HotpotQA R@5**|**MuSiQue R@2**|**MuSiQue R@5**|**2Wiki R@2**|**2Wiki R@5**|
> |-|-|-|-|-|-|-|
> |BM25|55.4|72.2|32.3|41.2|51.8|61.9|
> |Contriever|57.2|75.5|34.8|46.6|56.0|57.5|
> |GTR|59.4|73.3|37.4|47.9|60.2|67.0|
> |ColBERTv2|64.7|79.3|37.9|49.2|59.2|68.2|
> |RAPTOR|58.1|75.9|35.7|49.3|56.9|63.0|
> |Proposition|58.7|71.1|37.6|49.3|56.4|63.4|
> |GraphRAG (MS)|58.3|76.6|35.4|49.3|61.6|77.3|
> |LightRAG|38.8|54.7|31.4|43.9|51.0|67.9|
> |HippoRAG (Contriever)|59.0|76.2|39.1|52.1|71.5|89.5|
> |HippoRAG (ColBERTv2)|60.5|77.9|40.2|53.2|73.5|90.4|
> |SubgraphRAG|61.5|73.0|36.9|50.2|63.3|71.7|
> |G-retriever|53.3|65.5|38.8|45.1|60.8|67.5|
> |GFM-RAG|**78.3**|87.1|49.1|58.2|**90.8**|**95.6**|
> |**LP-RAG**|77.9|**89.6**|**53.6**|**61.8**|89.9|93.2|
>
> **Q&A performance**. The Table below reports Exact Match and F1. Here, LP-RAG is the best-performing method in all considered settings.
>
> |**Method**|**HotpotQA EM**|**HotpotQA F1**|**MuSiQue EM**|**MuSiQue F1**|**2Wiki EM**|**2Wiki F1**|
> |-|-|-|-|-|-|-|
> |None|30.4|42.8|12.5|24.1|31.0|39.0|
> |ColBERTv2|43.4|57.7|15.5|26.4|33.4|43.3|
> |GraphRAG (MS)|35.3|54.6|13.6|24.9|28.3|38.4|
> |LightRAG|36.8|48.3|18.1|27.5|45.1|49.5|
> |HippoRAG (ColBERTv2)|41.8|55.0|19.2|29.8|46.6|59.5|
> |GFM–RAG|51.6|66.9|30.2|40.4|69.8|77.7|
> |**LP–RAG**|**53.2**|**69.1**|**32.6**|**43.9**|**72.6**|**78.3**|
>
> > "W2 Scalability challenges...".
>
> To assess the efficiency of LP-RAG, we have run two additional sets of experiments. The first one compares against the best-performing graph-based competitors: LightRAG and NodeRAG, under the same hardware and setup to ensure a fair comparison. Our comparison covers indexing time (chunk generation + synthetic query generation + GNN training), storage usage (GNN parameters + size of resulting graph), average query latency, and average number of retrieved tokens. The results are below.
>
> |**Dataset**|**Size**|**Indexing time (min)**|**Storage (MB)**| **Query time (s)**|**Tokens**|
> |-|-|-|-|-|-|
> |||Light · Node · Ours|Light · Node · Ours|Light · Node · Ours|Light · Node · Ours|
> |HotpotQA|62K|7.9 · 5.1 · 4.7|21.2 · 10.7 · 10.3|5.8 · 4.2 · 3.6|5462 · 4019 · 3208|
> |MuSiQue| 58K|5.6 · 5.2 · 5.8|21.6 · 9.8 · 10.1|6.2 · 5.6 · 5.1 |5312 · 5269 · 3849|
> |MultiHop| 49K|5.9 · 5.3 · 4.8|19.7 · 10.4 · 9.7|7.4 · 5.9 · 5.1|6752 · 6032 · 3816|
> |RAG-QA|54K|6.1 · 5.8 · 6.2|26.8 · 14.0 · 12.1|5.6 · 4.8 · 4.2|6325 · 3370 · 2279|
> |ICLR|67K|10.3 · 8.6 · 7.9|38.7 · 26.9 · 15.7|7.9 · 5.4 · 5.7|6946 · 6214 · 2428|
>
> Overall, our method demonstrates equal or superior performance across all five datasets. In terms of query time, it is the fastest method on 4 out of 5 datasets, indicating that the overhead introduced by the GNN-based predictor is offset by the significantly lower number of retrieved tokens.
>
> For completeness, we have also considered larger datasets. In particular, we compare LP-RAG with NodeRAG using the same metrics on Arena Benchmarks (thousands of documents). The results are (Table 9 in the updated PDF):
>
> |**Dataset**|**Size**|**Indexing time (min)**|**Storage (MB)**|**Query time (s)**|**Tokens**|
> |-|-|--|-|-|-|
> |||Node Ours|Node Ours|Node Ours|Node Ours|
> |A. Writing|1.82M|13 · 183|157 · 175|5.40 · 6.32|3373 · 4219|
> |A. Tech|1.72M|14 · 165|139 · 168|6.74 · 7.98|3821 · 4863|
> |A. Science|1.43M|17 · 161|111 · 157|8.85 · 8.69|4284 · 4188|
> |A. Writing|0.93M|10 · 152|80 · 143|6.90 · 7.64|3448 · 4103|
>
> While LP-RAG introduces additional overhead in indexing time (training), its average query time remains comparable to that of NodeRAG. Given the strong empirical gains, we believe these results demonstrate both the feasibility and the practical potential of our approach on large-scale datasets.
>
> **(to be continued)**

---

> ### Author Response · Authors · 2025-11-21
> **Response: Part 2/2**
>
> > "W3-Q1 Transferability. How transferable is the approach? adapting to a new corpus typically requires regenerating synthetic queries and (re)training the link-prediction model rather than zero-shot transfer."
>
> Thanks for your question. We note that our approach is fully self-supervised. We do not use actual queries during training. Nonetheless, we have also run additional experiments regarding LP-RAG generalization.
>
> To do so, we consider two tasks: generalization to unseen documents within the same domain (dataset); cross-domain transferability. In the former, we set apart a collection of unseen documents during training, and incorporate only their chunk-graph for testing --- we did not retrain the GNN nor generate synthetic queries. In the latter, we train our model using one dataset, and assess how it performs on other ones (of a different domain). In particular, we trained LP-RAG on MuSiQue and assessed it on Hotpot and 2Wiki.
>
> The results for generalization to unseen documents are below. As we can see, LP-RAG remains the best model, outperforming GraphRAG, NodeRAG, and LightRAG.
>
> |**Method**|**HotpotQA-S**|**MuSiQue-S**|**MultiHop-S**|**RAG-QA-S**|**ICLR-S**|
> |------------|--------------|-------------|--------------|------------|----------|
> |NaiveRAG|52.00|42.00|38.00|54.00|64.00|
> |HyDe|64.00|68.00|66.00|62.00|70.00|
> |LightRAG|74.00|62.00|68.00|66.00|72.00|
> |NodeRAG|84.00|66.00|74.00|68.00|72.00|
> |LP-RAG|**86.00**|**82.00**|**84.00**|**82.00**|**84.00**|
>
> Regarding cross-domain transferability, the results (R@2 and R@5) are reported below. LP-RAG is the best model in 1/4 cases and the second-best in the remaining ones. It is important to note that GFM-RAG is pre-trained using a larger number of documents.
>
> |**Method**|**HotpotQA R@2**|**HotpotQA R@5**|**2Wiki R@2**|**2Wiki R@5**|
> |-|-|-|-|-|
> |BM25|55.4|72.2|51.8|61.9|
> |Contriever|57.2|75.5|56.0|57.5|
> |GTR|59.4|73.3|60.2|67.0|
> |ColBERTv2|64.7|79.3|59.2|68.2|
> |RAPTOR|58.1|75.9|56.9|63.0|
> |Proposition|58.7|71.1|56.4|63.4|
> |GraphRAG (MS)|58.3|76.6|61.6|77.3|
> |LightRAG|38.8|54.7|51.0|67.9|
> |HippoRAG (Contriever)|59.0|76.2|71.5|89.5|
> |HippoRAG (ColBERTv2)|60.5|77.9|73.5|90.4|
> |SubgraphRAG|61.5|73.0|63.3|71.7|
> |G-retriever|53.3|65.5|60.8|67.5|
> |GFM-RAG|**68.8**|**81.8**|**84.4**|89.6|
> |**LP-RAG**|65.4|78.3|75.9|**91.6**|
>
>
> _[1] GFM-RAG: graph foundation model for retrieval augmented generation. NeurIPS, 2025._
>
> ---
>
> Thank you again for the review and constructive feedback. We hope your concerns are satisfactorily addressed, and if so, would appreciate it if you could revisit your score to reflect the same. We are also committed to engage further if you have any more questions or suggestions.

---

### Official Review · Reviewer_JrmD · 2025-10-30

**Soundness:** 1
**Presentation:** 3
**Contribution:** 2
**Rating:** 2
**Confidence:** 4

**Summary:**

This paper proposes LP-RAG --- a link-prediction-based approach using GNNs for document retrieval-augmented generation (RAG). Empirical studies on nine multi-hop RAG benchmarks demonstrate the effectiveness of the proposed approach.

**Strengths:**

**S1.** The paper proposes a novel method for document RAG by leveraging GNNs for link prediction.

**S2.** Empirical studies span a large number of datasets.

**S3.** The paper is overall easy to follow.

**Weaknesses:**

**W1.** Most importantly, the empirical studies consider a very small subset of the corpus (<100 documents) for each dataset. Existing graph-based approaches for document RAG focus on open-domain question answering (QA) when it comes to multi-hop QA, where the corpus should span at least thousands of papers. Properly evaluating the performance and scalability of the proposed approach against the baselines requires sufficient studies under this setting.

**W2.** The paper fails to compare against several highly relevant works.

- Gutiérrez et al. HippoRAG: Neurobiologically Inspired Long-Term Memory for Large Language Models. An impactful work that performs knowledge graph extraction for document RAG. NeurIPS 2024.
- Gutiérrez et al. From RAG to Memory: Non-Parametric Continual Learning for Large Language Models. ICML 2025. An improved version of HippoRAG that incorporates chunk nodes.
- Luo et al. GFM-RAG: Graph Foundation Model for Retrieval Augmented Generation. NeurIPS 2025. GNN-based retrieval for document RAG.
- Alonso & Millidge. Mixture-of-PageRanks: Replacing Long-Context with Real-Time, Sparse GraphRAG. Also employs inter-chunk similarities for graph construction.
- Mavromatis & Karypis. GNN-RAG: Graph Neural Retrieval for Large Language Model Reasoning. This paper also leverages GNNs for graph retrieval.

Notably, many currently considered baselines are not designed to work well for multi-hop QA. E.g., GraphRAG tackles query-focused summarization, LightRAG is known to not perform well for multi-hop QA, etc.

**W3.** While the proposed approach involves training, the paper does not study the generalizability of the learned retrievers to datasets unseen during training.

**Questions:**

**Q1.** From section 3.2, it seems that the generated synthetic queries are all one-hop queries. Have you explored generating multi-hop synthetic queries?

---

> ### Author Response · Authors · 2025-11-21
> **Response: Part 1/2**
>
> Thank you for your review and for the opportunity to clarify our work. We have updated the PDF so that you can easily verify the changes. The revision includes comparisons against additional baselines on larger benchmarks and expanded analyses (e.g., efficiency and generalization to unseen documents). Below, we respond to each of the raised concerns.
>
> > “W1. the empirical studies consider a very small subset of the corpus (<100 documents) for each dataset...”
>
> Thanks for the opportunity to clarify this.
>
> We actually considered corpora with thousands of documents in our original submission. As detailed in Appendix B, Arena-Writing contains 11K documents and 2.6K questions; Arena-Science has 5.2K documents and 1.4K questions; Arena-Tech includes 8.4K documents and 2K questions; and Arena-Recreation comprises 4.3K documents and 2K questions.
>
> We also acknowledge that Table 1 contains smaller datasets. Our motivation was two-fold. First, the reference paper (NodeRAG, [1]) does not provide reproducible data splits, so we re-ran all baselines from scratch. Second, we used an open-source LLM both to reduce costs and to verify generalization across different LLM choices. To avoid confusion, we have added the suffix “-S” to small versions of these datasets (e.g., HotpotQA-S) in the revised manuscript (please, see updated Table 1).
>
> To further validate our method, we have now conducted additional experiments using the larger versions of MuSiQue, HotpotQA, and 2Wiki, following evaluation setup of previous works [e.g., GFM-RAG, 3]. We report the corresponding performance and scalability results in our next response.
>
> > "W2. The paper fails to compare against several highly relevant works."
>
> Thanks for the pointers to these works. As noted in our previous response, we conducted further experiments that compare our method with the following baselines: BM25, Contriever, GTR, ColBERTv2, RAPTOR, Proposition, GraphRAG, LightRAG, HippoRAG (Contriever), HippoRAG (ColBERTv2), SubgraphRAG, G-Retriever, and GFM-RAG. These experiments were carried out on MuSiQue, HotpotQA, and 2Wiki, following the evaluation protocol described in [HippoRAG, 2] and [GFM, 3]. To ensure a fair comparison, we used the same data splits, LLM configurations, and performance metrics as in those works.
>
> Below, we report performance regarding retrieval, Q&A, and computational performance in the single-step setting. Further implementation details are provided in Appendix B of the revised manuscript.
>
> **Retrieval performance**. Table below shows the recall@2 and recall@5 for LP-RAG and baselines. As we can observe, LP-RAG is the best model in 3/6 settings, and second best in the remaining ones.
>
> |**Method**|**HotpotQA R@2**|**HotpotQA R@5**|**MuSiQue R@2**|**MuSiQue R@5**|**2Wiki R@2**|**2Wiki R@5**|
> |-|-|-|-|-|-|-|
> |BM25|55.4|72.2|32.3|41.2|51.8|61.9|
> |Contriever|57.2|75.5|34.8|46.6|56.0|57.5|
> |GTR|59.4|73.3|37.4|47.9|60.2|67.0|
> |ColBERTv2|64.7|79.3|37.9|49.2|59.2|68.2|
> |RAPTOR|58.1|75.9|35.7|49.3|56.9|63.0|
> |Proposition|58.7|71.1|37.6|49.3|56.4|63.4|
> |GraphRAG (MS)|58.3|76.6|35.4|49.3|61.6|77.3|
> |LightRAG|38.8|54.7|31.4|43.9|51.0|67.9|
> |HippoRAG (Contriever)|59.0|76.2|39.1|52.1|71.5|89.5|
> |HippoRAG (ColBERTv2)|60.5|77.9|40.2|53.2|73.5|90.4|
> |SubgraphRAG|61.5|73.0|36.9|50.2|63.3|71.7|
> |G-retriever|53.3|65.5|38.8|45.1|60.8|67.5|
> |GFM-RAG|**78.3**|87.1|49.1|58.2|**90.8**|**95.6**|
> |**LP-RAG**|77.9|**89.6**|**53.6**|**61.8**|89.9|93.2|
>
>
> **Q&A performance**. The Table below reports Exact Match and F1. Here, LP-RAG is the best-performing method in all considered settings.
>
> |**Method**|**HotpotQA EM**|**HotpotQA F1**|**MuSiQue EM**|**MuSiQue F1**|**2Wiki EM**|**2Wiki F1**|
> |-|-|-|-|-|-|-|
> |None|30.4|42.8|12.5|24.1|31.0|39.0|
> |ColBERTv2|43.4|57.7|15.5|26.4|33.4|43.3|
> |GraphRAG (MS)|35.3|54.6|13.6|24.9|28.3|38.4|
> |LightRAG|36.8|48.3|18.1|27.5|45.1|49.5|
> |HippoRAG(ColBERTv2)|41.8|55.0|19.2|29.8|46.6|59.5|
> |GFM–RAG|51.6|66.9|30.2|40.4|69.8|77.7|
> |**LP–RAG**|**53.2**|**69.1**|**32.6**|**43.9**|**72.6**|**78.3**|
>
>
> **Computational aspect**. The table below reports the resource consumption of LP-RAG considering the controlled scenario reported in Section 4.1 of the updated PDF. LP-RAG offers equal or superior latency compared to the most modern graph-based RAG methods.
>
> |**Dataset**|**Size**|**Indexing time (min)**|**Storage (MB)**| **Query time (s)**|**Tokens**|
> |-|-|-|-|-|-|
> |||Light · Node · Ours|Light · Node · Ours|Light · Node · Ours|Light · Node · Ours|
> |HotpotQA-S|62K|7.9 · 5.1 · 4.7|21.2 · 10.7 · 10.3|5.8 · 4.2 · 3.6|5462 · 4019 · 3208|
> |MuSiQue-S| 58K|5.6 · 5.2 · 5.8|21.6 · 9.8 · 10.1|6.2 · 5.6 · 5.1 |5312 · 5269 · 3849|
> |MultiHop-S| 49K|5.9 · 5.3 · 4.8|19.7 · 10.4 · 9.7|7.4 · 5.9 · 5.1|6752 · 6032 · 3816|
> |RAG-QA-S|54K|6.1 · 5.8 · 6.2|26.8 · 14.0 · 12.1|5.6 · 4.8 · 4.2|6325 · 3370 · 2279|
> |ICLR-S|67K|10.3 · 8.6 · 7.9|38.7 · 26.9 · 15.7|7.9 · 5.4 · 5.7|6946 · 6214 · 2428|
>
> **(to be continued)**

---

> ### Author Response · Authors · 2025-11-21
> **Response: Part 2/2**
>
> > "Notably, many currently considered baselines are not designed to work well for multi-hop QA. E.g., GraphRAG tackles query-focused summarization, LightRAG is known to not perform well for multi-hop QA, etc."
>
> We initially followed the exact setup of NodeRAG, and thus we considered the same baselines. Nonetheless, as reported in the previous answer, we have additionally compared our method against multiple new baselines on larger datasets, following HippoRAG [2] and GFM-RAG [3].
>
> > "W3. While the proposed approach involves training, the paper does not study the generalizability of the learned retrievers to datasets unseen during training."
>
> While our method involves training, we note that it is done in a self-supervised manner via synthetic queries — we do not use real queries. This is different from other methods, like GFM-RAG, that conducts fine-tuning with actual queries.
>
> Nonetheless, we have run two sets of experiments to assess generalization/transferability performance. The first one evaluates model performance on unseen documents --- note that, in this setting, we do not use synthetic queries related to such documents. The second one evaluates transferability across datasets/domains.
>
> To evaluate generalization to unseen documents, we use the small-scale setting. After training LP-RAG, we collect 50 new documents for each dataset. For every new document, we generate its chunks and insert them into the inference graph, without retraining the GNN and without generating any additional synthetic queries. The accuracy results are shown in the table below. As we can see, LP-RAG remains the best-performing model even under this setup.
>
> |**Method**|**HotpotQA-S**|**MuSiQue-S**|**MultiHop-S**|**RAG-QA-S**|**ICLR-S**|
> |-|-|-|-|-|-|
> |NaiveRAG|52.00|42.00|38.00|54.00|64.00|
> |HyDe|64.00|68.00|66.00|62.00|70.00|
> |LightRAG|74.00|62.00|68.00|66.00|72.00|
> |NodeRAG|84.00|66.00|74.00|68.00|72.00|
> |**LP-RAG**|**86.00**|**82.00**|**84.00**|**82.00**|**84.00**|
>
> We now take the LP-RAG model trained on the MuSiQue dataset and evaluate it on HotpotQA and 2Wiki. The results are shown below. In this cross-dataset evaluation, LP-RAG is the best model in 1/4 cases and the second-best in the remaining ones. It is important to note that GFM-RAG is pre-trained using a larger number of documents.
>
> |**Method**|**HotpotQA R@2**|**HotpotQA R@5**|**2Wiki R@2**|**2Wiki R@5**|
> |-|-|-|-|-|
> |BM25|55.4|72.2|51.8|61.9|
> |Contriever|57.2|75.5|56.0|57.5|
> |GTR|59.4|73.3|60.2|67.0|
> |ColBERTv2|64.7|79.3|59.2|68.2|
> |RAPTOR|58.1|75.9|56.9|63.0|
> |Proposition|58.7|71.1|56.4|63.4|
> |GraphRAG (MS)|58.3|76.6|61.6|77.3|
> |LightRAG|38.8|54.7|51.0|67.9|
> |HippoRAG (Contriever)|59.0|76.2|71.5|89.5|
> |HippoRAG (ColBERTv2)|60.5|77.9|73.5|90.4|
> |SubgraphRAG|61.5|73.0|63.3|71.7|
> |G-retriever|53.3|65.5|60.8|67.5|
> |GFM-RAG|**68.8**|**81.8**|**84.4**|89.6|
> |**LP-RAG**|65.4|78.3|75.9|**91.6**|
>
> > "Q1. From section 3.2, it seems that the generated synthetic queries are all one-hop queries. Have you explored generating multi-hop synthetic queries?"
>
> Thanks for your question. We handle multi-hop synthetic queries by generating queries that span connected chunks, which strengthens the model’s ability to perform 2-, 3-, and 4-hop reasoning. In addition, we use multilayer GNNs for link prediction, allowing the model to further exploit multi-hop relational structure.
>
> References:
>
> _[1] NodeRAG: Structuring graph-based RAG with heterogeneous nodes. ArXiv e-prints, 2025._
>
> _[2] HippoRAG: Neurobiologically inspired long-term memory for large language models. NeurIPS, 2024._
>
> _[3] GFM-RAG: graph foundation model for retrieval augmented generation. NeurIPS, 2025._
>
>
> ---
>
> We hope your concerns have been satisfactorily addressed, and if so, would appreciate it if you could revisit your score to reflect the same. We are also committed to engaging further if you have any additional questions, concerns, or suggestions.

---

> ### Comment · Reviewer_JrmD · 2025-11-27
>
> Sorry for the late reply, and thank you for the detailed responses and extensive new results. They have largely addressed my concerns, and I’ve updated my ratings accordingly.
>
> One remaining suggestion is to avoid highlighting the small-corpus experiments in the main text for the next version. This setup is not standard for multi-hop RAG and may distract first-time readers from recognizing the paper's contributions. It would likely be clearer to move these results to the appendix. There is no need to be constrained by the particular NodeRAG paper.

---

> > ### Author Response · Authors · 2025-11-28
> >
> > Thank you for updating your assessment and for changing your recommendation in favor of acceptance. Your feedback was very helpful in improving the quality of the paper. We also appreciate your final suggestion regarding the small-corpus experiments. We agree that highlighting these results in the main text may mislead readers toward simpler, non-standard setups, and we will follow your suggestion.

---

### Official Review · Reviewer_m2DW · 2025-11-01

**Soundness:** 3
**Presentation:** 3
**Contribution:** 3
**Rating:** 6
**Confidence:** 4

**Summary:**

This paper proposes to build a graph of text and synthetic queries and then formulate retrieval for RAG as predicting link between incoming queries and text nodes. Specifically, the text nodes are connected as a kNN graph, and then one or several text nodes are used to generate synthetic queries that link to them. Several GNN-based link prediction methods are tested as the retrieval model.

**Strengths:**

- The idea of formulation retrieval as link prediction is interesting, and the usage of synthetic queries as training data is also intuitive and sound
- The experiment results overall is strong, showing the effectiveness of the method

**Weaknesses:**

- As claimed in the paper, the chunking strategy is also newly provided and is different from baselines. However, chunking is more considered as part of data process and should be kept identical among compared methods
- I didn't see in the paper what the selection criteria (threshold) is used for LP-RAG. The reproducibility can be further improved

**Questions:**

- Table 2, Tech, misses the boldfaced best scores, and LP-RAG is not strong as claimed in the text
- I wonder what a inference latency comparison looks like between the proposed GNN-based method and a typical dense retriever with FAISS
- The baseline can be enriched further. Also, I feel it is better to include a baseline that uses the same synthetic queries and their corresponding texts to fine-tune a standard dense retriever and make a comparison, which shows the contribution of the graph-based model

---

> ### Author Response · Authors · 2025-11-21
> **Response: Part 1/2**
>
> Thank you very much for the thoughtful and constructive review. We have updated the PDF to incorporate the suggestions from all reviewers. Below, we provide point-by-point responses to your comments.
>
> > "W1. [...] chunking is more considered as part of data process and should be kept identical among compared methods"
>
> Thanks for your suggestion. Several RAG methods come with specific ways to generate chunks and/or KGs, either using rigid schemas or Open IE. For instance, LightRAG [1] uses a fixed chunk size of 1200 tokens while NodeRAG [2] leverages LLMs for heterogenous KG construction. In our method (LP-RAG), the idea is to obtain chunks that capture atomic concepts — for which the range of possible questions (e.g., synthetic queries) is limited.
>
> Following prior works [e.g., 2], we tried to preserve as much as possible the original ideas and design choices of the baselines in our experiments. However, we agree that it would be interesting to observe the performance of the baselines under our chunks. The table below shows the macro accuracy for baselines using our chunk strategy.
>
> | **Method**              | **HotpotQA** | **MuSiQue** | **MultiHop** | **RAG-QA** | **ICLR**  |
> |-------------------------|--------------|-------------|--------------|------------|-----------|
> | NaiveRAG                | 18.33        | 15.56       | 16.67        | 12.78      | 11.11     |
> | HyDe                    | 25.46        | 21.19       | 36.50        | 32.67      | 37.69     |
> | LightRAG                | 71.25        | 60.62       | 64.25        | 61.20      | 66.62     |
> | NodeRAG                 | 83.31        | 62.29       | 70.64        | 67.05      | 74.43     |
> | **LP-RAG**              | **89.32**    | **89.67**   | **89.12**    | **87.89**  | **91.16** |
>
> As we can observe, when baselines use our chunks, accuracy drops significantly. We hypothesize that graphless approaches (e.g., NaiveRAG and HyDe) implicitly rely on larger chunks as they reduce information fragmentation and increase the likelihood of retrieving all relevant content within a single passage. This, in turn, lowers the need for multi-hop reasoning. In contrast, our approach tends to produce atomic chunks whose relationships are explicitly modeled through the graph structure and multi-hop GNN.
>
> We have included these results in Appendix D of the updated PDF.
>
> > "W2. I didn't see in the paper what the selection criteria (threshold) is used for LP-RAG. The reproducibility can be further improved."
>
> Thank you for pointing this out. We used $\tau=0.5$, which is a common choice in link-prediction settings, and we have added this information to Appendix B along with additional details. Finally, we also note that we will make our code publicly available for full reproducibility.
>
> > "Q1. Table 2, Tech, misses the boldfaced best scores, and LP-RAG is not strong as claimed in the text"
>
> We considered that LP-RAG and NodeRAG perform on par on Tech — that’s why we have not marked them in boldface. We have clarified this in the manuscript.
>
> > "Q2. I wonder what a inference latency comparison looks like between the proposed GNN-based method and a typical dense retriever with FAISS"
>
> Thanks for your question. Naturally, LP-RAG is expected to be slower than a simple Contriever+FAISS. For example, on HotpotQA, Contriever+FAISS achieves a retrieval latency of 0.0035 sec per query, whereas LP-RAG takes 0.235 sec. This includes the time to encode the query, insert it into the chunk graph, and run the GNN inference. That said, LP-RAG delivers substantially higher retrieval accuracy: on HotpotQA, it reaches 65.4 Recall@2 compared to Contriever’s 57.2.
>
> In our revised manuscript (please, see Tables 5 and 9 in the updated PDF), we have reported additional results in terms of indexing time (chunk generation + synthetic query generation + GNN training), storage usage (GNN parameters + size of resulting graph), average query latency, and average number of retrieved tokens. Overall, we found that LP-RAG performs comparably with other graph-based methods (e.g., LightRAG and NodeRAG). On larger datasets (e.g., Arena Benchmarks), while LP-RAG introduces additional overhead in indexing time, its average query time remains comparable to that of NodeRAG. Given the strong empirical gains, we believe these results demonstrate both the feasibility and the practical potential of our approach on large-scale datasets.
>
> **(To be continued...)**

---

> ### Author Response · Authors · 2025-11-21
> **Response: Part 2/2**
>
> > "Q3. The baseline can be enriched further..."
>
> Thanks for your suggestion to enrich baselines. This was indeed a recurring comment across reviewers. In response, we conducted an extensive comparison between LP-RAG and the baselines provided in [3], evaluating both retrieval performance and QA accuracy. To ensure fairness, we followed the exact same data splits, LLM configurations, and evaluation metrics used in [3].
>
> **Retrieval performance.** Table below shows the recall@2 and recall@5 for LP-RAG and baselines. As we can observe, LP-RAG is the best model in 3/6 settings, and second best in the remaining ones.
>
> | **Method**              | **HotpotQA R@2** | **HotpotQA R@5** | **MuSiQue R@2**  | **MuSiQue R@5**  | **2Wiki R@2**  | **2Wiki R@5**  |
> |-------------------------|------------------|------------------|------------------|------------------|----------------|----------------|
> | BM25                    | 55.4             | 72.2             | 32.3             | 41.2             | 51.8           | 61.9           |
> | Contriever              | 57.2             | 75.5             | 34.8             | 46.6             | 56.0           | 57.5           |
> | GTR                     | 59.4             | 73.3             | 37.4             | 47.9             | 60.2           | 67.0           |
> | ColBERTv2               | 64.7             | 79.3             | 37.9             | 49.2             | 59.2           | 68.2           |
> | RAPTOR                  | 58.1             | 75.9             | 35.7             | 49.3             | 56.9           | 63.0           |
> | Proposition             | 58.7             | 71.1             | 37.6             | 49.3             | 56.4           | 63.4           |
> | GraphRAG (MS)           | 58.3             | 76.6             | 35.4             | 49.3             | 61.6           | 77.3           |
> | LightRAG                | 38.8             | 54.7             | 31.4             | 43.9             | 51.0           | 67.9           |
> | HippoRAG (Contriever)   | 59.0             | 76.2             | 39.1             | 52.1             | 71.5           | 89.5           |
> | HippoRAG (ColBERTv2)    | 60.5             | 77.9             | 40.2             | 53.2             | 73.5           | 90.4           |
> | SubgraphRAG             | 61.5             | 73.0             | 36.9             | 50.2             | 63.3           | 71.7           |
> | G-retriever             | 53.3             | 65.5             | 38.8             | 45.1             | 60.8           | 67.5           |
> | GFM-RAG                 | **78.3**         | 87.1             | 49.1             | 58.2             | **90.8**       | **95.6**       |
> | **LP-RAG**              | 77.9             | **89.6**         | **53.6**         | **61.8**         | 89.9           | 93.2           |
>
> **Q&A performance.** The Table below reports Exact Match and F1. Here, LP-RAG is the best-performing method in all considered settings.
>
> | **Method**              | **HotpotQA EM** | **HotpotQA F1** | **MuSiQue EM** | **MuSiQue F1** | **2Wiki EM** | **2Wiki F1** |
> |-------------------------|------------------|------------------|------------------|------------------|---------------|--------------|
> | None                    | 30.4             | 42.8             | 12.5             | 24.1             | 31.0          | 39.0         |
> | ColBERTv2               | 43.4             | 57.7             | 15.5             | 26.4             | 33.4          | 43.3         |
> | GraphRAG (MS)           | 35.3             | 54.6             | 13.6             | 24.9             | 28.3          | 38.4         |
> | LightRAG                | 36.8             | 48.3             | 18.1             | 27.5             | 45.1          | 49.5         |
> | HippoRAG (ColBERTv2)    | 41.8             | 55.0             | 19.2             | 29.8             | 46.6          | 59.5         |
> | GFM–RAG             | 51.6             | 66.9             | 30.2             | 40.4             | 69.8          | 77.7         |
> | **LP–RAG**              | **53.2**         | **69.1**         | **32.6**         | **43.9**         | **72.6**      | **78.3**     |
>
> These additional results are reported in Section 4.3 of the updated PDF.
>
> _[1] LightRAG: Simple and fast retrieval augmented generation. EMNLP, 2025._
>
> _[2] NodeRAG: Structuring graph-based RAG with heterogeneous nodes. ArXiv e-prints, 2025._
>
> _[3] GFM-RAG: Graph foundation model for retrieval augmented generation. NeurIPS, 2025._
>
> ---
>
> Thank you again for the constructive feedback. We hope our response has fully addressed your concerns, and we would greatly appreciate your strengthened support for this work. If you have any further questions or suggestions, please let us know.

---

> > ### Comment · Reviewer_m2DW · 2025-11-25
> >
> > Thanks for the detailed response with the additional experiments! I do not have further concerns on the paper.
> >
> > I think my initial overall rating of a 6 reflects my final judgement and I would like to vote for an accept. Good luck!

---

> > > ### Author Response · Authors · 2025-11-25
> > > **Thank you**
> > >
> > > Thank you for acknowledging our rebuttal and for appreciating our work. We are also glad to hear that you are voting for acceptance. If any additional concerns arise in the remaining days, we would be happy to discuss them.

---

### Meta-Review · Area_Chair_WNVN · 2026-01-07

**Summary:**

The paper proposes LP-RAG, a framework that reformulates the retrieval component of Retrieval-Augmented Generation (RAG) as an inductive link prediction task. The method decomposes documents into atomic chunks using an LLM, generates synthetic queries for these chunks, and constructs a chunk-query graph. A GNN (specifically a Neural Common Neighbor model) is trained on this graph to predict links between user queries and chunks. The approach aims to improve retrieval precision and reduce token consumption by explicitly learning query-chunk compatibility via synthetic supervision.

**Reviewer Concerns:**

Addressed Concerns:
1. Baselines: The authors successfully addressed the lack of comparisons against state-of-the-art methods by adding GFM-RAG, HippoRAG, and GraphRAG baselines on larger datasets (MuSiQue, HotpotQA, 2Wiki) during the rebuttal.

2. Chunking Isolation: The authors demonstrated that their performance gains are not solely due to the atomic chunking strategy, as applying this chunking to baselines like NaiveRAG degraded their performance.

Outstanding Concerns:
1. High Indexing Cost & Complexity: Reviewer M6Nt and 9Z9c raised significant concerns regarding scalability and efficiency. While the authors provided data, the results confirm a massive overhead in indexing time compared to baselines (e.g., 183 minutes for LP-RAG vs. 13 minutes for NodeRAG on the Arena-Writing dataset).

2. Practicality of Pipeline: The workflow requires three heavy steps: LLM-based chunking, LLM-based synthetic query generation for every chunk, and GNN training. For dynamic corpora or real-world applications requiring frequent updates, this preprocessing burden is likely prohibitive compared to the marginal gains over simpler graph or dense retrieval methods.

**Reviewer Scores:**

Reviewer JrmD (2 -> 6): Explicitly raised their score, noting the "extensive new results" addressed their concerns.

---

### Decision · Program_Chairs · 2026-01-26

Reject